# MAVRL: Learning Reward Functions from Multiple Feedback Types with Amortized Variational Inference

Raphaël Baur [1]   Yannick Metz [2]   Maria Gkoulta [2]   Mennatallah El-Assady [2,1]
Giorgia Ramponi [3]   Thomas Kleine Buening [2,1]

## Abstract

Reward learning typically relies on a single feedback type or combines multiple feedback types using manually weighted loss terms. Currently, it remains unclear how to jointly learn reward functions from heterogeneous feedback types such as demonstrations, comparisons, ratings, and stops that provide qualitatively different signals. We address this challenge by formulating reward learning from multiple feedback types as Bayesian inference over a shared latent reward function, where each feedback type contributes information through an explicit likelihood. We introduce a scalable amortized variational inference approach that learns a shared reward encoder and feedback-specific likelihood decoders and is trained by optimizing a single evidence lower bound. Our approach avoids reducing feedback to a common intermediate representation and eliminates the need for manual loss balancing. Across discrete and continuous-control benchmarks, we show that jointly inferred reward posteriors outperform single-type baselines, exploit complementary information across feedback types, and yield policies that are more robust to environment perturbations. The inferred reward uncertainty further provides interpretable signals for analyzing model confidence and consistency across feedback types.

## 1. Introduction

Designing reward functions that faithfully capture desired behavior is notoriously difficult. Even in well-specified environments, subtle preferences, safety considerations, and trade-offs are hard to encode by hand, and small misspecifications can lead to unintended or unsafe behavior (Abouelazm et al., 2024; Amodei et al., 2016; Gershman & Niv, 2015; Hendrycks et al., 2021; Knox et al., 2023). This challenge has motivated a broad line of work on reward learning, where reward functions are inferred from human feedback rather than specified explicitly. Such approaches are appealing because they allow domain experts to communicate intent through judgments or interventions that are often easier to provide than a complete formal specification.

Human feedback about behavior, however, comes in many forms. Previous work has studied learning from demonstrations (Chan & van der Schaar, 2021; Ng et al., 2000), comparisons or preferences (Christiano et al., 2017; Wirth et al., 2016), scalar ratings (Knox & Stone, 2009), rankings (Brown et al., 2019; Myers et al., 2021), and interventions such as corrections or emergency stops (Ghosal et al., 2023; Hadfield-Menell et al., 2017; Losey et al., 2022), among others. Each feedback type provides only partial information about the underlying reward function: demonstrations offer sparse coverage of the state-action space and constrain rewards only along the expert behaviors; comparisons provide relative information about returns without fully specifying trade-offs beyond the compared alternatives; ratings only provide ordinal information about trajectories while discarding preference strength; and stops typically indicate unacceptable behavior without specifying what would have been optimal instead. As a result, relying on any single feedback type can leave aspects of the reward undetermined.

These limitations highlight the need for learning from multiple feedback types, as different modalities can provide complementary information that resolves ambiguities left by any single source. Moreover, feedback types differ in availability, cost, and informativeness, making it important to understand how they relate to one another, how much information they provide, where they overlap, and when they conflict.

Despite their complementary potential, learning reward functions jointly from heterogeneous feedback types remains challenging. Existing approaches either train separate reward models for each feedback type and combine them

[1]ETH AI Center, ETH Zurich, Zurich, Switzerland [2]Department of Computer Science, ETH Zurich, Zurich, Switzerland [3]Department of Informatics, University of Zurich, Zurich, Switzerland. Correspondence to: Raphaël Baur <raphael.baur@ai.ethz.ch>.

*Proceedings of the 43rd International Conference on Machine Learning*, Seoul, South Korea. PMLR 306, 2026. Copyright 2026 by the author(s).

post hoc (Ibarz et al., 2018; Macuglia et al., 2025; Metz et al., 2025), or focus on a narrow subset of modalities, typically demonstrations and comparisons (Bıyık et al., 2022). Both strategies introduce significant challenges: post hoc aggregation raises questions about how to reconcile reward scales and uncertainties across feedback types, while collapsing diverse feedback into a single intermediate representation (such as preferences) can obscure modality-specific information and is not applicable to all forms of feedback. As a result, despite their complementary potential, jointly learning reward functions from multiple types of feedback remains difficult in practice.

A more principled perspective is to view each feedback type as a probabilistic observation of a shared latent reward function. Under this formulation, learning from multiple feedback types naturally corresponds to Bayesian inference, where each feedback modality contributes information through its likelihood. Several frameworks formalize human feedback in this manner, for example, by modeling feedback as reward-rational choices from (possibly implicit) choice sets (Jeon et al., 2020). This likelihood-based formulation provides a conceptually unified treatment of heterogeneous feedback and makes explicit how different feedback types relate to the same underlying reward. However, exact Bayesian inference in this setting is generally intractable due to the need to marginalize over both feedback realizations and reward functions.

To address these challenges, we introduce a scalable amortized variational inference approach for learning reward functions from multiple feedback types. Building on prior work on scalable Bayesian inverse reinforcement learning (Chan & van der Schaar, 2021), our method learns a shared variational reward representation together with feedback-specific likelihood models and is trained by optimizing a single evidence lower bound.

**Contributions** Concretely, our contributions are as follows:

- We introduce a unified Bayesian formulation for learning reward functions from multiple feedback types, where each feedback modality contributes information through an explicit likelihood and no manual loss balancing is required (Section 4).
- We propose a scalable amortized variational inference algorithm that jointly learns a shared reward representation together with feedback-specific likelihood models (Section 5).
- We empirically demonstrate that jointly learning from multiple feedback types exploits complementary information, improves reward recovery and policy robustness, and yields interpretable reward uncertainty across a range of reinforcement learning benchmarks (Section 6)[1].

## 2. Related Work

**Reward Learning.** Reward learning seeks to infer reward functions from human feedback when explicit reward specification is impractical (Abbeel & Ng, 2004; Christiano et al., 2017; Ng et al., 2000). Early work focused on inverse reinforcement learning (IRL) from demonstrations, assuming expert trajectories arise from (approximately) optimal behavior under an unknown reward (Abbeel & Ng, 2004; Ng et al., 2000). More recently, preference-based reinforcement learning has gained prominence, particularly through applications in language modeling, where humans compare agent trajectory segments and reward models are trained using the Bradley-Terry model or related probabilistic choice formulations (Christiano et al., 2017; Ouyang et al., 2022).

Beyond demonstrations and preferences, a variety of other feedback types have been explored, including corrections (Bajcsy et al., 2017; Losey et al., 2022), rankings (Brown et al., 2020; Myers et al., 2021), and emergency stops (Hadfield-Menell et al., 2017). Each feedback paradigm introduces its own modeling assumptions and loss functions, and is typically studied in isolation. A unifying perspective was proposed by Jeon et al. (2020), who showed that many feedback types can be interpreted as reward-rational choices from (possibly implicit) choice sets. While this framework provides a common probabilistic interpretation of feedback, it does not by itself yield a scalable method for jointly learning from heterogeneous feedback sources.

**Approximate Inference for Bayesian IRL.** Closely related to our work is the literature on scalable Bayesian IRL. Bayesian IRL poses reward learning from demonstrations as posterior inference, but early methods relied on MCMC or other sampling-based inference methods, limiting their applicability to small tasks (Ramachandran & Amir, 2007; Rothkopf & Dimitrakakis, 2011).

To address this, Chan & van der Schaar (2021) proposed AVRIL, which applies amortized variational inference to Bayesian IRL by jointly learning a variational reward encoder and a demonstration likelihood decoder. This formulation enables efficient posterior inference without repeatedly solving a reinforcement learning problem in an inner loop and has been shown to scale to high-dimensional control as well as transformer-based reward models in language modeling (Cai et al., 2025).

Other recent work has applied variational inference to preference learning to capture user-specific reward variation, but with a different objective than learning a shared reward function from multi-type feedback (Poddar et al., 2024). We build directly on the AVRIL framework by replacing its sin-

---

[1]Code and trained models are available at `https://github.com/rabaur/mavrl`.

gle demonstration likelihood with a set of feedback-specific likelihood models, while maintaining a single shared variational posterior over reward functions. This enables joint amortized inference from several feedback types without collapsing them into a common surrogate objective.

**Multi-Type Feedback.** Compared to single-type reward learning, relatively little work has studied learning from multiple types of human feedback. Most existing efforts focus on demonstrations and preferences, and use demonstrations primarily as an initialization step before applying preference-based learning to further refine the policy and reward estimates (Bıyık et al., 2022; Ibarz et al., 2018; Macuglia et al., 2025; Palan et al., 2019).

A small number of approaches attempt to incorporate more than two feedback types within a single learning procedure. Mehta & Losey (2024) integrate demonstrations, corrections, and preferences using the reward-rational choice framework in a robotics setting, but ultimately combine modalities through additive loss terms whose relative influence is fixed by design choices (e.g., sampling rates).

Recent benchmark suites and evaluation platforms emphasize the practical relevance of heterogeneous feedback (Metz et al., 2023; Yuan et al., 2024), and existing large-scale studies evaluate combinations of feedback types using ensemble-style approaches (Metz et al., 2025). However, these approaches do not perform joint inference over a single reward function and instead rely on heuristics to reconcile different reward scales and uncertainties across feedback types.

In contrast, our approach performs joint Bayesian inference over a shared reward function from multi-type feedback by integrating feedback-specific likelihoods within a single variational objective.

## 3. Preliminaries

We consider Markov Decision Processes (MDPs) $\mathcal{M} = (\mathcal{S}, \mathcal{A}, T, R^*, \gamma)$ with state space $\mathcal{S}$, action space $\mathcal{A}$, transition dynamics $T(s' \mid s, a)$, ground-truth reward function $R^* : \mathcal{S} \times \mathcal{A} \to \mathbb{R}$, and discount factor $\gamma \in [0, 1)$. Both $\mathcal{S}$ and $\mathcal{A}$ may be discrete or continuous, and we do not assume access to the transition dynamics. Moreover, the reward function $R^*$ is assumed to be unobserved.

The agent interacts with the MDP via a (possibly stochastic) stationary policy $\pi(a|s)$, which induces a distribution over trajectories $\xi = (s_0, a_0, s_1, a_1, \dots)$ in the MDP. We denote by $\Pi$ the space of all stationary policies and with $\Xi$ the space of all trajectories.[2] Given a reward function $R$, the

return of a trajectory $\xi$ is $R(\xi) = \sum_{t=0}^{\infty} \gamma^t R(s_t, a_t)$, and the expected return of a policy $\pi$ is $R(\pi) = \mathbb{E}_{\xi \sim \pi}[R(\xi)]$. For a given reward function $R$, we let $Q_R^*(s, a) = \sup_{\pi \in \Pi} \mathbb{E}_{\xi \sim \pi}[R(\xi) \mid s_0 = s, a_0 = a]$ denote the optimal action-value function.

**Bayesian Learning from Multi-Type Feedback.** Given a set of multi-type human feedback $\mathcal{D}$, we are interested in learning a reward function. Concretely, we consider $\mathcal{D} = \{\mathcal{D}^{(m)}\}_{m=1}^M$, where $\mathcal{D}^{(m)}$ corresponds to feedback of type $m$ (e.g., preferences).

Taking the Bayesian perspective, our goal is to infer a posterior distribution over the reward function given this data. Following this, we treat the reward function $R$ as a latent variable and consider

$$p(R \mid \mathcal{D}) = \frac{p(\mathcal{D} \mid R)p(R)}{\int p(\mathcal{D} \mid R')p(R')\, \mathrm{d}R'}, \tag{1}$$

where $p(R)$ is a prior over reward functions and $p(\mathcal{D} \mid R)$ is the likelihood of observing the multi-type feedback $\mathcal{D}$ under $R$. Crucially, the likelihood of observed feedback conditional on $R$ factorizes in a useful manner. As the observations are conditionally independent given the reward function, we can express the joint likelihood as $p(\mathcal{D} \mid R) = \prod_m p(\mathcal{D}^{(m)} \mid R)$. Hence, the multi-type feedback posterior satisfies $p(R \mid \mathcal{D}) \propto \prod_m p(\mathcal{D}^{(m)} \mid R)p(R)$.

**Amortized Variational Inference.** Although elegant, equation (1) is generally doubly intractable: due to integrating over feedback choices in the likelihood and over all reward functions in the denominator. *Variational inference* (VI) addresses this intractability by approximating the posterior with a simpler *variational* distribution $q_\theta(R) \approx p(R \mid \mathcal{D})$, whose parameters $\theta$ are learned by maximizing the evidence lower bound (ELBO):

$$\mathbb{E}_{R \sim q_\theta(\cdot)}[\log p(\mathcal{D} \mid R)] - D_{\mathrm{KL}}(q_\theta(R) \,\|\, p(R)). \tag{2}$$

The first term maximizes the expected data likelihood, and the second softly regularizes $q_\theta(R)$ toward the prior $p(R)$.

The *Variational Autoencoder* (VAE) framework (Kingma & Welling, 2014) made variational inference broadly applicable and scalable by representing the variational distribution $q_\theta(z \mid x)$ as a neural network *encoder* that maps observations $x$ to latent variables $z$, thereby *amortizing* posterior inference across data points. In addition, VAEs jointly learn a likelihood model $p_\phi(y \mid z)$, the *decoder*, which reconstructs the observed response $y$ from the latent representation $z$.[3] This is enabled through the *reparameterization trick*, which expresses sampling as a deterministic transformation $z = g_\theta(\epsilon, x)$ with $\epsilon \sim p(\cdot)$, allowing gradients to flow through the encoder.

---

[2] All definitions and results extend straightforwardly to finite-horizon MDPs by replacing the infinite discounted sum with a finite-horizon return. We adopt the infinite-horizon discounted formulation for notational convenience.

[3] Typically, in VAEs, the objective is to learn meaningful latent representations $z$ through a reconstruction task, where $x = y$.

## 4. Feedback-Specific Likelihood Models

We define probabilistic likelihood models for each feedback type, which together induce the joint data likelihood $p(\mathcal{D} \mid R)$ in the Bayesian reward learning objective (Eq. 1) and therefore determine the variational objective optimized by our algorithm. In this paper, we focus on *preferences* (pairwise comparisons), *demonstrations*, *ratings*, and *stops*, but the proposed framework and algorithm apply to any feedback type for which a probabilistic likelihood can be specified. Importantly, our method is agnostic to the particular choice of likelihood model and does not rely on shared intermediate representations or manual loss balancing across feedback types.

**Preferences (PDRS).** We here employ the commonly used Bradley-Terry model under which a trajectory $\xi_1$ is preferred over $\xi_2$ under the reward function $R$ with probability:

$$p(\xi_1 \succ \xi_2 \mid R) = \frac{\exp\big(\beta R(\xi_1)\big)}{\exp\big(\beta R(\xi_1)\big) + \exp\big(\beta R(\xi_2)\big)}.$$

This formulation naturally extends to comparisons between trajectory segments, as in Christiano et al. (2017). The inverse temperature parameter $\beta$ controls the stochasticity of the preference judgments.

**Demonstrations (PDRS).** Expert demonstrations represent another commonly used form of feedback. Ramachandran & Amir (2007) and subsequent work, e.g., Chan & van der Schaar (2021); Rothkopf & Dimitrakakis (2011), model expert demonstrations through a Boltzmann-rational policy $\pi(a \mid s, R) \propto \exp\big(\beta Q_R^*(s, a)\big)$. We follow their modeling so that the likelihood of an expert trajectory $\xi$ under the reward function $R$ is given by

$$p(\xi \mid R) = \prod_{(s,a) \in \xi} \frac{\exp\big(\beta Q_R^*(s, a)\big)}{\sum_{b \in \mathcal{A}} \exp\big(\beta Q_R^*(s, b)\big)},$$

where the inverse temperature $\beta > 0$ controls the degree of expert optimality.

**Ratings (PDRS).** Ratings provide ordinal feedback in which a human assigns a discrete score to a trajectory reflecting its perceived quality, analogous to Likert-scale judgments (Likert, 1932). We model this form of feedback using a standard ordinal regression framework, given by the ordered logit model (McCullagh, 1980).

We assume that each trajectory induces an unobserved latent utility reflecting its quality under the reward function $R$, corrupted by stochastic judgment noise. The reported rating $y \in \{1, \ldots, K\}$ is generated by discretizing this latent utility via an ordered set of cutpoints $\psi_0 = -\infty < \psi_1 < \cdots < \psi_K = +\infty$, such that a trajectory $\xi$ receives rating $y = k$ whenever its latent utility falls between $\psi_{k-1}$ and $\psi_k$.

Under this model, the likelihood of observing rating $y = k$ for trajectory $\xi$ is

$$p(y = k \mid \xi, R) = F(\psi_k - R(\xi)) - F(\psi_{k-1} - R(\xi)),$$

where $F$ denotes the logistic cumulative distribution function. Importantly, this feedback modality is not equivalent to direct regression on the reward value, as it only provides ordinal information, but no absolute judgment. The cutpoints $\{\psi_k\}$ need not be evenly spaced, reflecting the fact that humans typically would not apply uniform or linear thresholds when mapping perceived quality to discrete ratings.

**Stops (PDRS).** Stop signals capture situations in which a human supervisor intervenes to terminate an agent's behavior once its performance has degraded beyond an acceptable level. Such feedback is ubiquitous in practice, for example, as safety stops in robotics or human-in-the-loop control, yet has received little attention as a learning signal for reward or policy learning. Our framework naturally accommodates stop feedback by modeling it through an explicit likelihood over termination times.

Let $\tau_{\text{stop}}$ be the (random) time step at which the user intervenes. We model $\tau_{\text{stop}}$ using a discrete-time hazard model in which the instantaneous hazard at time $\tau$ depends on the accumulated suboptimality of the trajectory up to that point. Concretely, we define the instantaneous suboptimality of action $a$ in state $s$ under reward $R$ as $\Delta_R(s, a) = \max_{b \in \mathcal{A}} Q_R^*(s, b) - Q_R^*(s, a)$, and introduce a backward-looking discount factor $\rho \in (0, 1]$ that discounts earlier deviations. The resulting hazard function is given by

$$h_R^{\lambda, \rho}(\xi, \tau) = 1 - \exp\Big( - \lambda \sum_{t=1}^{\tau} \rho^{\tau - t} \Delta_R(s_t, a_t)\Big),$$

Here, the larger $\lambda > 0$ is, the more unforgiving the expert. We obtain the likelihood of observing a stop at time $\tau$ as the geometric distribution:

$$p(\tau_{\text{stop}} = \tau \mid R) = \underbrace{\Bigg[\prod_{t=1}^{\tau-1}(1 - h_R^{\lambda, \rho}(\xi, t))\Bigg]}_{\text{no stop up until } \tau} h_R^{\lambda, \rho}(\xi, \tau). \quad (3)$$

If no stop occurs within the segment, the observation is right-censored.

**Additional Feedback Types.** Other feedback types studied in the literature include *rankings* (Brown et al., 2019; Myers et al., 2021), *corrections* (Losey et al., 2022), and other forms of *human intervention* (Jeon et al., 2020). While we do not explicitly model these here, feedback-specific

---

**Algorithm 1** (`MAVRL`) Multi-Feedback Amortized Variational Reward Learning

---

**input:** Multi-type feedback $\{\mathcal{D}^{(m)}\}_{m=1}^{M}$, hyperparameters $\lambda_{\text{KL}}, \lambda_{\text{TD}}$, learning rate $\eta$
**while** *not converged* **do**

> Sample mini-batches for all feedback types: $\mathcal{B} \leftarrow \{\texttt{GetBatch}(\mathcal{D}^{(m)})\}_{m=1}^{M}$
> Encode all transitions: $(\boldsymbol{\mu}, \boldsymbol{\sigma}^2) \leftarrow q_\theta(\mathcal{B})$
> Reparameterize rewards: $\boldsymbol{R} \leftarrow \boldsymbol{\mu} + \boldsymbol{\sigma} \odot \boldsymbol{\epsilon}, \quad \boldsymbol{\epsilon} \sim \mathcal{N}(\boldsymbol{0}, \boldsymbol{I})$
> Evaluate $Q$-values: $\boldsymbol{Q} \leftarrow Q_\phi(\mathcal{B})$
> Compute feedback-specific negative log-likelihoods: $\{\mathcal{L}_{\text{NLL}}^{(m)}\}_{m=1}^{M} \leftarrow \{-\log p_\psi^{(m)}(\mathcal{B}^{(m)} \mid \boldsymbol{R}, \boldsymbol{Q})\}_{m=1}^{M}$
> $\mathcal{L}_{\text{KL}} \leftarrow D_{\text{KL}}(q_\theta \,\|\, p)$, with prior $p = \mathcal{N}(\boldsymbol{0}, \boldsymbol{I})$
> $\mathcal{L}_{\text{TD}} \leftarrow \text{TD}(\boldsymbol{\mu}, \boldsymbol{\sigma}, \boldsymbol{Q})$
> Aggregate losses: $\mathcal{L}_{\text{total}} \leftarrow \sum_{m=1}^{M} \mathcal{L}_{\text{NLL}}^{(m)} + \lambda_{\text{KL}} \mathcal{L}_{\text{KL}} + \lambda_{\text{TD}} \mathcal{L}_{\text{TD}}$
> Update parameters: $(\theta, \phi) \leftarrow (\theta, \phi) - \eta \nabla_{(\theta, \phi)} \mathcal{L}_{\text{total}}$

**end**
**return** reward encoder $q_\theta$, action-value model $Q_\phi$

---

likelihoods can be derived analogously. Crucially, any such likelihood can be incorporated into our framework and algorithm without additional structural changes. We provide implementation details on feedback simulation in Section A.2.

## 5. `MAVRL`: Multi-Feedback Amortized Variational Reward Learning

We now present `MAVRL`, a general learning algorithm for Bayesian reward inference from multiple feedback types. The algorithm (Algorithm 1) instantiates the Bayesian objective introduced in Section 3 using amortized variational inference and directly leverages the feedback-specific likelihood models defined in Section 4.

### 5.1. Objective

We aim to learn a probabilistic reward model together with an auxiliary action-value function that supports feedback types defined in terms of state-action values. Specifically, we parameterize a conditional reward distribution

$$q_\theta(R \mid s, a) = \mathcal{N}\big(R; \mu_\theta(s, a), \sigma_\theta^2(s, a)\big),$$

where $\mu_\theta$ and $\sigma_\theta^2$ are parameterized as neural networks that map tuples $(s, a)$ to a distribution over local reward values. Sampling from this encoder induces a distribution over trajectory returns, which is used to evaluate feedback likelihoods.

In addition, we learn an auxiliary action-value function $Q_\phi(s, a)$, which is required by feedback types that depend on value estimates, as well as parameters $\psi$ associated with feedback-specific likelihood models when applicable, e.g., the cutpoints of the rating model. Crucially, the reward encoder itself is agnostic to the type and structure of feedback; all feedback-specific semantics are captured entirely by the corresponding likelihood functions.

Learning follows directly from the evidence lower bound (ELBO) of the Bayesian reward learning problem. Given

feedback datasets $\mathcal{D}^{(m)}{}_{m=1}^{M}$, where each $\mathcal{D}^{(m)}$ contains observations of feedback type $m$, the unified `MAVRL` objective to be maximized is given by

$$\mathcal{L}_{\text{MAVRL}}(\theta, \phi, \psi) =$$
$$\sum_{m=1}^{M} \mathbb{E}_{y \sim \mathcal{D}^{(m)}} \Big[ \mathbb{E}_{R \sim q_\theta} \big[ \log p_\psi^{(m)}(y \mid R, Q_\phi) \big] \Big]$$
$$- \lambda_{\text{KL}} \, D_{\text{KL}}(q_\theta(R) \,\|\, p(R)) + \lambda_{\text{TD}} \, \mathcal{L}_{\text{TD}}(\theta, \phi). \quad (4)$$

Here, $y$ denotes a generic feedback observation whose structure depends on the feedback type, such as a trajectory, a comparison, a scalar rating, or a termination time. Each feedback type contributes a likelihood term to the objective without requiring manual weighting or staged optimization.

The final term $\mathcal{L}_{\text{TD}}(\theta, \phi)$ enforces consistency between the inferred reward distribution and the auxiliary action-value function. Following Chan & van der Schaar (2021), we include a temporal-difference (TD) regularization term that encourages rewards predicted by the encoder to agree with the one-step Bellman differences implied by $Q_\phi$.

To define this term, we leverage the trajectories associated with the observed feedback. Each feedback observation $o$ is grounded in one or more trajectories, from which we extract state-action transitions. Let $\tilde{\tau} = (s, a, s', a')$ denote such a transition tuple, where $a'$ is the action taken in state $s'$. The corresponding TD target is defined as

$$\delta_\phi(\tilde{\tau}) := Q_\phi(s, a) - \gamma Q_\phi(s', a').$$

We penalize deviations between this TD target and the reward predicted by the encoder by maximizing its log-likelihood under the encoder distribution,

$$\mathcal{L}_{\text{TD}}(\theta, \phi) = \mathbb{E}_{\tilde{\tau} \sim \mathcal{D}_{\text{traj}}} \Big[ \log \mathcal{N}\big(\delta_\phi(\tilde{\tau}); \mu_\theta(s, a), \sigma_\theta^2(s, a)\big) \Big],$$

where $\mathcal{D}_{\text{traj}}$ denotes the set of transitions extracted from the trajectories underlying the observed feedback.

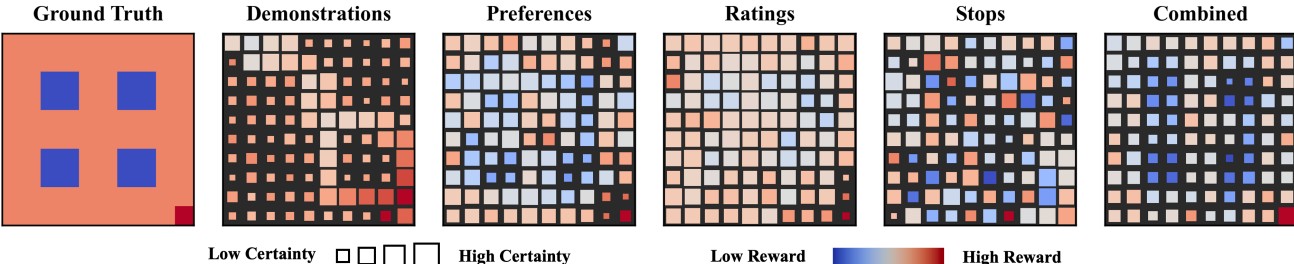

*Figure 1.* Visualizations of inferred reward functions from 2 demonstrations, 256 pairwise comparisons, or 128 ratings on a $10 \times 10$ grid_trap environment. The final column shows the result obtained when combining all feedback modalities.

## 5.2. Properties of `MAVRL`

**Extensibility.** A central property of `MAVRL` is that it can use any form of human feedback for which a likelihood can be specified. Each feedback type contributes to the objective only through its likelihood term $p^{(m)}(y \mid R, Q_\phi)$. Thus, adding a new feedback modality requires only defining its likelihood, without changing the reward encoder, auxiliary value function, or optimization.

**Unified Objective Without Cross-Modal Loss Balancing.** `MAVRL` does not assume that different feedback modalities share a common intermediate representation or supervision signal besides the reward function. Feedback-specific likelihoods relate observations to the reward via different statistical relationships, such as action probabilities, return comparisons, ordinal thresholds, or cumulative regret. Because all feedback appears as log-likelihood terms in a single variational objective, the relative influence of each modality is determined by the data and feedback models, not by hand-tuned cross-modal weights. Thus, `MAVRL` removes the need to choose how much each feedback type should weigh in the reward estimator: For instance, with $n_d = 1$ demonstration and $n_p = 25$ preferences and known noise parameters, their relative weights in a combined loss or post-hoc ensemble are *arbitrary* in general MDPs. `MAVRL` avoids this issue entirely.

Two parameter categories remain: *Likelihood-specific noise parameters* (e.g., Bradley–Terry temperature $\beta$, rating cutpoints $\{\psi_k\}$, and stop hazard parameters $\lambda, \rho$) model annotator behavior and are grounded in established human-feedback models; they can optionally be treated as latent variables and inferred under behaviorally plausible priors. *Regularization coefficients* $\lambda_{\text{KL}}$ and $\lambda_{\text{TD}}$ control posterior regularization and reward–Q-value consistency; we tune them via standard model selection, and performance is stable over a broad range of values, with Q-value-based feedback (demonstrations and stops) requiring non-zero $\lambda_{\text{TD}}$.

**Order-Invariant and Asynchronous Training.** The training objective is agnostic to the order, frequency, and interleaving of feedback modalities. Mini-batches from each

type-specific feedback dataset are sampled independently and can be combined arbitrarily during optimization.

## 6. Experiments

We evaluate `MAVRL` across a range of environments and feedback configurations to answer three questions:

(i) how different feedback types qualitatively complement one another when learning reward functions (Section 6.1),

(ii) how this complementarity translates into downstream policy performance and reward fidelity (Section 6.2),

(iii) whether rewards inferred from multi-type feedback lead to more robust behavior under environment perturbations or reward misspecification (Section 6.3).

Our experiments span both tabular grid-world domains and continuous-control benchmarks, and consider demonstrations, pairwise comparisons, ratings, and stop feedback both in isolation and in combination.

Unless otherwise stated, all methods are trained under a fixed feedback budget per environment, with identical numbers of feedback samples allocated to each modality to enable fair comparison. Let $n_p$, $n_d$, $n_r$, and $n_s$ denote the numbers of preferences, demonstrations, ratings, and stop signals, respectively. In the grid-world environments, we set $n_p = n_r = 64$, $n_d = 1$, and $n_s = 256$. In Acrobot-v1 and CartPole-v1, we set $n_p = n_r = 256$, $n_d = 4$, and $n_s = 64$. In LunarLander-v3, we set $n_p = n_r = n_s = 256$, and $n_d = 32$.

### 6.1. Feedback Types Complement One Another

In Figure 1, we visualize reward estimates learned with `MAVRL` based on different individual feedback types and their combination in a grid-world environment. We report both the learned mean reward estimate and its corresponding uncertainty, quantified by the variance. Additional examples can be found in Section F.

We find that each feedback type induces a characteristic

pattern in the inferred reward and uncertainty estimates. Demonstrations yield low-uncertainty reward estimates along expert trajectories, but leave large regions of the state space underdetermined. Pairwise preferences provide broader coverage, but can assign a high reward to frequently visited intermediate states that are not globally optimal. Ratings reliably identify the goal state, but offer limited information about the surrounding reward landscape. Stop feedback strongly constrains unsafe or low-reward regions, while providing relatively little guidance on desirable behavior beyond avoidance.

Illustrating the complementary nature of different types of feedback, these limitations are mitigated when they are combined. Our reward model, trained on combined feedback, can reconstruct the original ground-truth reward function with high fidelity and reliably identify both high- and low-reward states.

## 6.2. Combining Feedback Improves Policy Performance and Reward Identification

We evaluate the downstream policy performance of different feedback combinations across a range of grid-world and continuous-control environments in Table 1. For each combination, we optimize a policy w.r.t. the learned reward estimate and report the policy's total return w.r.t. the ground-truth reward function.

All results are averaged over 10 runs, and the returns are linearly scaled per environment so that 100 corresponds to an approximately optimal policy trained on the ground-truth reward and 0 corresponds to the return of a uniformly random policy. Hence, values exceeding 100 indicate that the learned policy outperforms the policy trained on the ground-truth reward.

While these normalized metrics capture downstream policy performance, they do not globally reflect reward recovery. To assess this, we also reports EPIC distance (Gleave et al., 2021), which is zero for reward functions that are equivalent up to potential-based shaping, positive scaling, and constant shifts. We report EPIC distance only for tabular environments, where exact reward comparisons are feasible.

**Combining all feedback types yields the strongest overall performance.** Overall, PDRS performs strongly across environments, achieving either the best or near-best performance in four out of six environments, with the exception of Acrobot-v1 and LunarLander-v3. This supports our central hypothesis that MAVRL is capable of leveraging complementary information captured in different feedback types.

**No single-type baseline dominates across all environments.** Perhaps unsurprisingly, demonstrations (PDRS) excel in sparse reward settings, such as grid_sparse (95.0) and

LunarLander-v3 (115.7), while ratings (PDRS) perform best in CartPole-v1 (100) among the single feedback type approaches. Preferences (PDRS) show relatively modest performance as a standalone modality, yet contribute substantially when combined with others, consistent with prior findings that pairwise comparisons require large quantities to achieve competitive performance (Christiano et al., 2017).

**Stop feedback complements other feedback.** In several environments, combinations involving stop feedback (PDRS, PDRS, PDRS) generally outperform their non-stop counterparts. For instance, in grid_sparse, PDRS (100.0) improves over PDRS alone (95.0), and PDRS (70.6) similarly improves over PDRS (58.8). This suggests that stop signals provide valuable information about the suboptimality of trajectories that is otherwise difficult to obtain from other feedback.

**EPIC distance reveals reward recovery quality beyond policy performance.** In the three tabular environments where EPIC is computed PDRS attains the lowest (grid_cliff: 0.481; grid_trap: 0.520) or second-lowest (grid_sparse: 0.383) EPIC distance. Discrepancies between EPIC and performance only occur on grid_sparse, where PDRS reaches the lowest EPIC (0.380) but only intermediate policy performance (82.3), indicating that faithful reward recovery and high downstream performance, while not strictly equivalent, are jointly achieved by combining all feedback modalities.

**MAVRL is both effective and efficient relative to non-variational baselines.** Table 2 compares MAVRL against two natural alternatives: MCMC, a non-amortized Bayesian reward learning method representing the gold standard for posterior inference, and Post-Hoc Reward Averaging, a non-variational heuristic that combines independently trained reward estimates through ensemble averaging (cf. Metz et al., 2025).

Against MCMC, MAVRL achieves comparable or superior performance at roughly $30\times$ lower compute on grid environments and, unlike MCMC, does not require an inner-loop MDP solve, making it tractable on continuous-control settings where MCMC is not. Against Post-Hoc Averaging, MAVRL substantially outperforms naïve ensembling across all environments, indicating that independently trained reward models conflate signals with different scales and shaping and miss the cross-modal complementarity that joint inference captures.

**Gains persist under matched feedback budgets.** To check whether the improvements from combining feedback types reflect genuine complementarity rather than larger total supervision, we conducted an equal-budget ablation with fixed total samples allocated across modalities via Bayesian optimization (Appendix C, Table 4). On continuous-control

| | PDRS | PDRS | PDRS | PDRS | PDRS | PDRS | PDRS | PDRS | PDRS | PDRS | PDRS |
|---|---|---|---|---|---|---|---|---|---|---|---|
| *Performance (normalized return)* ↑ | | | | | | | | | | | |
| grid_cliff | 59.2 | 80.4 | 41.1 | 84.1 | 80.1 | 91.9 | 77.0 | 99.7 | 99.2 | 72.1 | 100.0 |
| grid_sparse | 80.0 | 95.0 | 58.8 | 90.0 | 90.0 | 82.3 | 85.0 | 94.1 | 100.0 | 70.6 | 100.0 |
| grid_trap | 78.3 | 46.5 | 60.4 | 46.1 | 77.1 | 90.0 | 86.2 | 82.0 | 84.9 | 64.1 | 95.3 |
| Acrobot-v1 | 74.4 | 99.7 | 99.2 | 99.5 | 99.0 | 84.3 | 92.4 | 98.9 | 99.9 | 98.3 | 97.9 |
| CartPole-v1 | 92.2 | 97.4 | 100.0 | 96.5 | 100.0 | 100.0 | 95.5 | 100.0 | 100.0 | 100.0 | 100.0 |
| LunarLander-v3 | 47.5 | 115.7 | 56.6 | 69.3 | 75.6 | 73.3 | 53.2 | 105.6 | 104.5 | 64.0 | 93.5 |
| *EPIC distance* ↓ | | | | | | | | | | | |
| grid_cliff | 0.614 | 0.651 | 0.593 | 0.593 | 0.533 | 0.553 | 0.606 | 0.588 | 0.626 | 0.638 | 0.481 |
| grid_sparse | 0.445 | 0.456 | 0.470 | 0.413 | 0.453 | 0.380 | 0.423 | 0.520 | 0.442 | 0.427 | 0.383 |
| grid_trap | 0.627 | 0.708 | 0.622 | 0.719 | 0.562 | 0.585 | 0.532 | 0.609 | 0.608 | 0.522 | 0.520 |

*Table 1.* Normalized mean returns and mean EPIC distances ($n = 10$) for different combinations of feedback modalities across environments. Returns (top, higher is better) are normalized such that 100 corresponds to the performance of an approximately optimal policy trained on the ground-truth reward and 0 corresponds to the performance of a uniformly random policy. EPIC distances (bottom, lower is better) measure the divergence between learned and ground-truth rewards. Cells shaded in dark gray indicate the best result for the corresponding environment, and cells shaded in light gray indicate the next best result outside this margin.

benchmarks, combinations consistently match or outperform the strongest single modality, with PDRS among the top performers. On the tabular grids, a single informative modality at full budget often beats fragmented multi-modality allocations, and pairs such as PDRS or PDRS emerge as the strongest combinations. Overall, combining feedback yields genuine cross-modal complementarity, with the benefit of including a given modality depending on its informativeness in the target environment.

| | Post-Hoc Avg | MCMC | | MAVRL | |
|---|---|---|---|---|---|
| Environment | Perf. | Perf. | W.-Time | Perf. | W.-Time |
| grid_sparse | 94.1 | 100.0 | 154.4 s | 100.0 | 4.5 s |
| grid_cliff | 54.9 | 100.0 | 156.0 s | 100.0 | 4.9 s |
| grid_trap | 47.3 | 93.0 | 155.0 s | 95.3 | 4.6 s |
| LunarLander-v3 | 59.9 | – | – | 93.5 | 46.5 s |

*Table 2.* Comparison of MAVRL with non-variational baselines. Performance (Perf.) is normalized return averaged over $n{=}10$ seeds (higher is better); wall-time (W.-Time) is per reward-model training run on identical hardware. Best per row in dark gray, second-best in light gray.

### 6.3. Rewards Learned from Multi-Type Feedback are More Robust

We evaluate the robustness of reward functions inferred using MAVRL along two complementary axes: (i) robustness to shifts in environment dynamics at deployment, and (ii) robustness to noise and misspecification in the feedback used at training time. In both settings, reward models are trained once in the nominal configuration, then held fixed while policies are retrained without further reward adaptation, isolating the effect of reward quality from policy

optimization. We compare individual feedback modalities, their combination (PDRS), and a behavioral cloning baseline (IMIT). Full experimental details and results for all feedback combinations are provided in Section E.

**Multi-type feedback degrades more gracefully under dynamics perturbations.** We perturb the transition dynamics of each environment after reward learning: increased action stochasticity in grid_cliff and grid_trap, structural changes to the system dynamics in Acrobot-v1, and shifts in gravity and wind in LunarLander-v3. Figure 2 shows the performance under these perturbations. In grid-environments, policies trained on rewards inferred from multiple feedback types degrade more gracefully under perturbations than those trained from individual feedback modalities or imitation alone. While downstream performance of combined feedback still degrades more gracefully than the imitation baseline, single-modality performance of, e.g., PDRS, exceeds it, consistent with the unperturbed setting. In the grid-world setting, increasing action stochasticity causes imitation and single-feedback reward models such as PDRS, PDRS, and PDRS to deteriorate rapidly, whereas rewards learned from combined feedback maintain higher mean returns. A similar pattern is observed in Acrobot-v1, where structural changes to the system dynamics disproportionately affect the imitated policy and the single-feedback reward models, while combined feedback yields consistently more stable performance across perturbation levels. In LunarLander-v3, perturbations to gravity and wind severely degrade the performance of IMIT. In contrast, MAVRL, especially when trained with multi-type feedback (PDRS) and demonstration feedback (PDRS), remains significantly more robust under increasingly challenging dynamics.

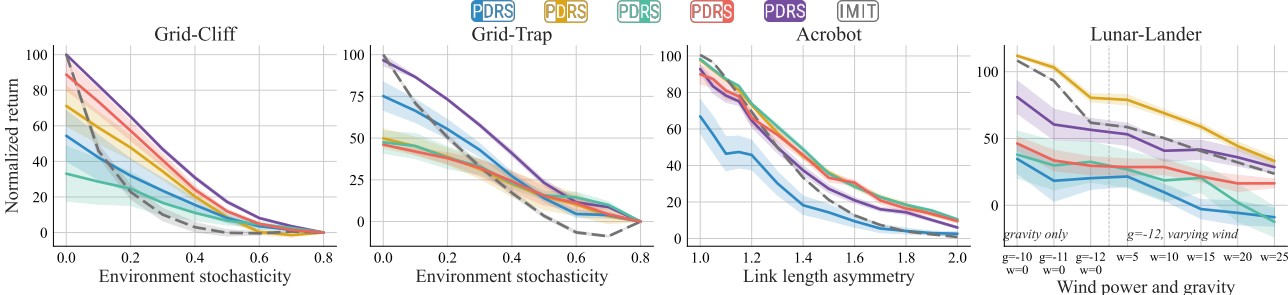

*Figure 2.* Normalized mean returns ($n = 10$) of policies trained on rewards inferred by each method under three dynamics perturbation scenarios. Reward models and baselines are trained in the unperturbed setting and remain fixed throughout the variations. Error bars denote standard error. (left) Increasing environmental stochasticity in grid_cliff and grid_trap. (middle) Increasing ratio between pendulum handle lengths in Acrobot-v1. (right) Increasing gravity and wind-power in LunarLander-v3.

| | grid_cliff | | | grid_sparse | | |
|---|---|---|---|---|---|---|
| | Base. | Misspec. | Rat. (↑) | Base. | Misspec. | Rat. (↑) |
| PD**RS** | 59.2 | 46.9 ± 14.5 | 0.79× | 80.0 | 20.0 ± 13.3 | 0.25× |
| PD**RS** | 80.4 | 15.0 ± 10.7 | 0.19× | 95.0 | 9.9 ± 10.0 | 0.10× |
| PD**RS** | 41.1 | 29.6 ± 11.7 | 0.72× | 58.8 | 30.0 ± 15.3 | 0.51× |
| PD**RS** | 84.1 | 29.0 ± 11.8 | 0.35× | 90.0 | 20.0 ± 13.3 | 0.22× |
| **P**DRS | 100.0 | 90.3 ± 8.7 | 0.90× | 100.0 | 90.0 ± 10.0 | 0.90× |
| P**D**RS | 100.0 | 38.1 ± 13.3 | 0.38× | 100.0 | 50.0 ± 16.7 | 0.50× |
| PD**R**S | 100.0 | 100.0 ± 0.0 | 1.00× | 100.0 | 100.0 ± 0.0 | 1.00× |
| PDR**S** | 100.0 | 100.0 ± 0.0 | 1.00× | 100.0 | 90.0 ± 10.0 | 0.90× |
| **PDRS** | 100.0 | 43.2 ± 13.2 | 0.43× | 100.0 | 50.0 ± 16.7 | 0.50× |

*Table 3.* Misspecification robustness ($n$=10; grid_trap in App. D). Top: single-modality models; bottom: MAVRL with full feedback set. Each row corrupts the data-side parameter of the named channel while the model assumes the well-specified value. Corrupted modality is blurred; e.g., **PD**RS denotes misspecified preferences. *Base.* = well-specified normalized return; *Misspec.* = mean±SEM; *Rat.* = Misspec./Base. Best per column in dark gray, second-best in light gray.

**Multi-type feedback compensates for most misspecified modalities.** Next, we test whether the multi-type advantage extends to corruption of the feedback signal itself, under the same fixed allocation feedback budget inmavrl Section 6.2. We apply four misspecifications, individually or jointly: underestimated labeler rationality for preferences ($\beta_{\mathrm{pref}}$) and demonstrations ($\beta_{\mathrm{demo}}$), additive Gaussian noise on rating utilities, and miscalibrated stop propensity. Results for grid_cliff and grid_sparse are shown in Table 3; the same pattern holds on grid_trap (Table 5).

Two observations stand out. First, for three of the four corruptions (preferences, ratings, and stops) PD**RS** retains $\geq 90\%$ of its well-specified performance, while the corresponding single-modality baselines collapse to ratios as low as 0.10. This suggests a strong compensation effect: Joint inference over multiple feedback types absorbs corruption from one source rather than propagating it. Second, the exception is demonstrations: corrupting them drives PD**RS** to

ratios of 0.38–0.50, and corrupting *all* four channels simultaneously reaches roughly the same level (0.43–0.50). The worst-case degradation of PD**RS** is therefore bottlenecked by the demonstration channel rather than compounded across channels. Overall, the multi-type benefit extends to upstream corruption in three of four feedback channels, with demonstrations remaining the structural weak point.

## 7. Conclusion

We framed reward learning from multiple feedback types as Bayesian inference over a shared latent reward function, where each feedback modality contributes through an explicit likelihood, and proposed MAVRL, a scalable amortized variational inference algorithm that optimizes a single unified objective. Empirically, we found that different feedback types induce distinct reward and uncertainty structures, and jointly inferring rewards from multiple modalities can exploit their complementary strengths, improving reward recovery, downstream policy performance, and robustness.

**Limitations.** Our evaluation is limited to simulated environments, which, while allowing controlled analysis of different feedback types, do not capture the full complexity of real-world tasks. In addition, we rely on synthetically generating feedback, and applying MAVRL to real human feedback, which may exhibit systematic biases, inconsistencies, or context-dependent judgments, remains an important direction for future work. Finally, while our framework infers reward uncertainty, we do not yet exploit this uncertainty for guiding feedback collection, leaving active learning of heterogeneous feedback as a promising avenue for future work.

## Acknowledgments

This research was primarily supported by the ETH AI Center through an ETH AI Center doctoral fellowship to Raphaël Baur and an ETH AI Center postdoctoral fellowship to Thomas Kleine Buening.

## Impact Statement

This paper presents work whose goal is to advance the field of machine learning. There are many potential societal consequences of our work, none of which we feel must be specifically highlighted here.

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

# A. Implementation Details

## A.1. Model Architecture

The reward encoder $q_\theta$ and Q-value estimator $Q_\phi$ were implemented as two-layer MLPs with Leaky ReLU activations. For grid environments, we used learning rate $5 \times 10^{-4}$, batch size 32, and state-only rewards $R(s)$. For control tasks (CartPole-v1, Acrobot-v1, LunarLander-v3), we used learning rate $10^{-4}$, batch size 128, and state-action rewards $R(s, a)$. All models were trained using AdamW with gradient clipping (max norm 1.0) (Loshchilov & Hutter, 2019). The reward encoder parameterizes a Gaussian posterior with a standard normal prior, trained via the reparameterization trick. Expert reference policies were trained with StableBaseline3's DQN implementation (all except LunarLander-v3) and PPO implementation (LunarLander-v3) (Raffin et al., 2021).

**Hyperparameter optimization.** For each environment and feedback combination, hyperparameters are tuned with Optuna's multivariate TPE sampler (`constant_liar=True`, 20 startup trials) (Akiba et al., 2019). Each trial trains the model under a sampled configuration across multiple seeds and is scored by the chosen validation metric. The joint search space covers:

- Loss weights $\lambda_{\text{TD}}$ and $\lambda_{\text{KL}}$, log-uniform in $[10^{-3}, 2.0]$;
- Reward-encoder width: $16, 32, 64$ hidden units per layer for grids, $64, 128, 256$ for control tasks;
- Reward-model optimizer: learning rate (log-uniform $[10^{-5}, 10^{-3}]$), batch size ($16, 32, 64, 128$ for grids, $32, 64, 128, 256$ for control);
- For LunarLander-v3, the PPO retraining hyperparameters (learning rate, clip range, entropy/value coefficients, GAE-$\lambda$, gradient-norm bound, discount, rollout length, mini-batches, epochs) are searched jointly with the reward-model parameters.

We run this tuning under two feedback-allocation modes:

- *Equal-budget (Dirichlet) mode*: a fixed total feedback budget is split across the active modalities via a flat-Dirichlet allocation suggested by Optuna. Per-modality caps (e.g., $\leq 16 / \leq 32$ demonstrations for grids/control, since demonstration likelihoods saturate quickly) are enforced by clipping the Dirichlet proportions and redistributing the excess mass.
- *Fixed-allocation mode*: each modality is assigned a prescribed sample count (`FIXED_SAMPLE_COUNTS`) and only the optimization hyperparameters above are searched. This is used to characterize each modality's contribution at a useful, comparable budget without confounding it with allocation choices.

The same tuning protocol (sampler, search space, seeds-per-trial, validation metric) is applied uniformly to all methods within an environment so reported numbers reflect best-of-search performance under matched compute.

Model and training code was implemented with PyTorch (Paszke et al., 2019).

## A.2. Feedback Simulation

We simulate human feedback from trajectories collected by rolling out Boltzmann-rational policies with temperature parameter $\beta_{\text{traj}}$. Below we describe the generation process for each feedback type.

**Preferences.** We extract random segments of fixed length $L$ from collected trajectories. For each pair of segments $(\xi_1, \xi_2)$, we compute their normalized returns $\bar{R}(\xi_i) = R(\xi_i)/L$ and generate a preference according to the Bradley-Terry model:

$$p(\xi_1 \succ \xi_2) = \sigma\left(\beta_{\text{pref}} \cdot (\bar{R}(\xi_1) - \bar{R}(\xi_2))\right),$$

where $\sigma$ denotes the logistic sigmoid and $\beta_{\text{pref}}$ controls preference rationality. The observed preference is sampled as a Bernoulli random variable with this probability. We choose $\beta_{\text{pref}} = 5.0$, segment length $L = 10$, for grids and $L = 32$ for all other environments, trajectory rationality $\beta_{\text{traj}} = 0.0$ for grids (uniform policy), 5.0 for all other environments.

**Demonstrations.** Expert demonstrations are generated by rolling out a Boltzmann-rational policy $\pi(a \mid s) \propto \exp(\beta_{\text{demo}} \cdot Q^*(s, a))$, where $Q^*$ denotes the optimal Q-function. The demonstration rationality $\beta_{\text{demo}}$ controls the degree of expert optimality. We choose $\beta_{\text{demo}} = 10.0$ for grids, $\beta_{\text{demo}} = 5.0$ for all remaining environments.

**Ratings.** We extract random segments from trajectories and compute their normalized returns. Cutpoints $\mu_k$ are placed at the $(100 \cdot k/K)$-th percentiles of the per-step segment returns. Ratings are then sampled stochastically under a cumulative-

logit model, $P(y = k \mid R) = \sigma(\mu_k - R) - \sigma(\mu_{k-1} - R)$, which yields approximately balanced categories. Typical parameters: $K = 5$ categories, segment length $L \in \{10, 32\}$ (grids, all other environments), trajectory rationality $\beta_{\text{traj}} \in \{0.0, 1.0, 5.0\}$ (grids, Acrobot-v1 and CartPole-v1, LunarLander-v3 respectively).

**Stops.** We simulate stop feedback using a discrete-time hazard model based on cumulative regret. For a trajectory segment, we compute the instantaneous regret at each time step as $\Delta_t = \max_b Q^*(s_t, b) - Q^*(s_t, a_t)$ and maintain a discounted cumulative regret:

$$\mathcal{R}_t = \rho \mathcal{R}_{t-1} + \Delta_t,$$

where $\rho \in [0, 1]$ is the regret discount factor controlling how quickly past suboptimality is "forgotten". The hazard rate (probability of stopping at time $t$ given no prior stop) is:

$$h_t = 1 - \exp(-\lambda \mathcal{R}_t).$$

The sensitivity parameter $\lambda$ is calibrated from the data as $\lambda = c/\mathcal{R}_{\text{ref}}$, where $c$ is a scaling constant and $\mathcal{R}_{\text{ref}}$ is a reference regret level (typically the 50th percentile of maximum cumulative regrets across segments). Larger $c$ yields more aggressive stopping behavior. Stop times are sampled sequentially: at each time step $t$, we sample a stop event with probability $h_t$. If no stop occurs within the segment, the observation is right-censored. Parameters: $c = 1.0$, regret discount $\rho = 0.1$, reference percentile $= 50\%$, segment length $L \in \{10, 32\}$ (grids, all remaining environments), trajectory rationality $\beta_{\text{traj}} \in \{0.0, 1.0\}$.

## B. Hardware and Computational Resources

Due to the large number of evaluated configurations in Section 6, individual reward model training and evaluation runs were distributed on a SLURM-managed institutional cluster, with each worker allocated 4 CPU cores and 8 GB of RAM. Due to the small size of the reward encoder and Q-value estimator networks, no GPU acceleration was required. For the final experiments, the total compute consumption amounted to approximately $2.7 \times 10^5$ CPU-hours.

We note that an *individual* reward model training run takes approximately 2 minutes for the grid environments and about 15 minutes for all other environments on a single machine, with most of the computation time spent on retraining a policy using the inferred reward function for evaluation.

## C. Equal Budget Results

For each environment we fix a single cumulative feedback budget $N$ shared across all 11 modality subsets considered in the equal-budget table: each of the four feedback types alone (PDRS, PDRS, PDRS, PDRS), all six pairwise combinations, and the full combination of all four (PDRS). The budget is $N = 64$ samples for the three tabular grids (grid_cliff, grid_sparse, grid_trap), Acrobot-v1, and CartPole-v1, and $N = 256$ samples for LunarLander-v3. For every (environment, modality subset) pair, we run a separate Tree-structured Parzen Estimator (TPE) search that jointly tunes (i) the proportion of $N$ allocated to each active feedback type and (ii) the reward-model training hyperparameters (encoder size, learning rate, batch size, importance-weighting toggle, KL weight, TD weight); for non-tabular environments the search additionally tunes the downstream PPO retraining hyperparameters. Each trial trains the reward model under 20 random seeds for tabular environments and 10 for non-tabular environments, and is scored by the per-seed mean of normalized discounted return (tabular) or mean episodic return (non-tabular); we run between 125 and 200 trials per study. The best-trial reward-model checkpoints are the ones used in all downstream transfer and misspecification experiments reported in this paper.

## D. Full Misspecification Results

We considered four misspecification variations, applied either individually or jointly: (i) underestimated labeler noisiness for preferences ($\beta_{\text{data}} = 0.01$, $\beta_{\text{model}} = 5$) and demonstrations ($\beta_{\text{data}} = 0.1$, $\beta_{\text{model}} = 10$), where the model assumes a near-rational labeler while the data is generated under highly noisy labeling; (ii) noisy ratings (Gaussian noise $\sigma_{\text{data}} = 2$ added to segment utilities before label assignment, while the model assumes noise-free utilities); and (iii) misspecified stop propensity ($c_{\text{data}} = 0.1$, $c_{\text{model}} = 2$). In all cases the reward model is trained with the same well-specified parameters used in the main table, so the unperturbed baseline is unchanged.

## E. Additional Transfer Results and Detailed Perturbation Description

Transfer performance results for all feedback combinations are shown in Figure 3. We now summarize the environmental perturbations used to evaluate the robustness of downstream policy performance under systematic changes in environment

| | PDRS | PDRS | PDRS | PDRS | PDRS | PDRS | PDRS | PDRS | PDRS | PDRS | PDRS |
|---|---|---|---|---|---|---|---|---|---|---|---|
| *Performance (normalized return)* ↑ | | | | | | | | | | | |
| grid_cliff | 59.0 ±10.4 | 45.4 ±9.5 | 61.6 ±11.6 | 16.9 ±6.6 | 67.5 ±10.2 | 64.9 ±11.1 | 45.8 ±11.7 | 99.1 ±0.4 | 33.6 ±8.2 | 41.2 ±12.1 | 61.9 ±12.2 |
| grid_sparse | 70.0 ±10.5 | 60.7 ±11.0 | 64.7 ±12.0 | 15.0 ±8.2 | 60.0 ±11.2 | 58.8 ±12.3 | 15.0 ±8.2 | 88.2 ±8.1 | 60.0 ±11.2 | 58.8 ±12.3 | 52.9 ±12.5 |
| grid_trap | 74.5 ±6.1 | 45.0 ±1.1 | 59.2 ±5.0 | 41.7 ±3.0 | 55.9 ±5.1 | 75.5 ±6.9 | 69.8 ±5.8 | 55.5 ±5.2 | 41.3 ±3.4 | 64.1 ±7.1 | 46.3 ±0.2 |
| Acrobot-v1 | 98.3 ±0.9 | 99.6 ±0.3 | 99.8 ±0.1 | 99.8 ±0.2 | 99.6 ±0.1 | 97.5 ±0.7 | 98.4 ±0.6 | 99.2 ±0.2 | 99.8 ±0.2 | 99.6 ±0.1 | 99.8 ±0.1 |
| CartPole-v1 | 100.0 ±0.0 | 100.0 ±0.0 | 100.0 ±0.0 | 100.0 ±0.0 | 100.0 ±0.0 | 100.0 ±0.0 | 97.8 ±1.3 | 100.0 ±0.0 | 100.0 ±0.0 | 100.0 ±0.0 | 100.0 ±0.0 |
| LunarLander-v3 | 47.5 ±4.8 | 115.7 ±0.2 | 56.6 ±10.2 | 69.3 ±4.8 | 94.0 ±13.4 | 41.2 ±14.2 | 66.1 ±5.7 | 101.4 ±4.3 | 90.1 ±5.3 | 51.4 ±11.1 | 112.4 ±2.0 |
| *EPIC distance* ↓ | | | | | | | | | | | |
| grid_cliff | 0.612 ±0.018 | 0.660 ±0.023 | 0.617 ±0.012 | 0.705 ±0.007 | 0.620 ±0.023 | 0.604 ±0.014 | 0.625 ±0.019 | 0.591 ±0.006 | 0.684 ±0.007 | 0.630 ±0.015 | 0.631 ±0.018 |
| grid_sparse | 0.424 ±0.048 | 0.558 ±0.023 | 0.475 ±0.047 | 0.698 ±0.017 | 0.465 ±0.071 | 0.555 ±0.039 | 0.660 ±0.014 | 0.519 ±0.009 | 0.578 ±0.039 | 0.510 ±0.048 | 0.565 ±0.042 |
| grid_trap | 0.633 ±0.006 | 0.705 ±0.003 | 0.577 ±0.008 | 0.720 ±0.003 | 0.584 ±0.013 | 0.633 ±0.005 | 0.606 ±0.006 | 0.607 ±0.010 | 0.706 ±0.007 | 0.590 ±0.011 | 0.478 ±0.015 |

*Table 4.* Normalized mean returns and mean EPIC distances ($n = 10$) for different combinations of feedback modalities across environments **under equal feedback budget**. Returns (top, higher is better) are normalized such that 100 corresponds to the performance of an approximately optimal policy trained on the ground-truth reward and 0 corresponds to the performance of a uniformly random policy. EPIC distances (bottom, lower is better) measure the divergence between learned and ground-truth rewards. Cells shaded in dark gray indicate the best result for the corresponding environment (within 1% for returns), and cells shaded in light gray indicate the next best result outside this margin.

| | Grid-Cliff | | | Grid-Sparse | | | Grid-Trap | | |
|---|---|---|---|---|---|---|---|---|---|
| | Base. | Misspec. | Ratio (↑) | Base. | Misspec. | Ratio (↑) | Base. | Misspec. | Ratio (↑) |
| PDRS | 59.2 | 46.9 ± 14.5 | 0.79× | 80.0 | 20.0 ± 13.3 | 0.25× | 78.3 | 43.0 ± 4.3 | 0.55× |
| PDRS | 80.4 | 15.0 ± 10.7 | 0.19× | 95.0 | 9.9 ± 10.0 | 0.10× | 46.5 | 43.0 ± 8.4 | 0.93× |
| PDRS | 41.1 | 29.6 ± 11.7 | 0.72× | 58.8 | 30.0 ± 15.3 | 0.51× | 60.4 | 35.5 ± 6.8 | 0.59× |
| PDRS | 84.1 | 29.0 ± 11.8 | 0.35× | 90.0 | 20.0 ± 13.3 | 0.22× | 46.1 | 46.1 ± 0.0 | 1.00× |
| PDRS | 100.0 | 90.3 ± 8.7 | 0.90× | 100.0 | 90.0 ± 10.0 | 0.90× | 95.3 | 91.9 ± 5.8 | 0.96× |
| PDRS | 100.0 | 38.1 ± 13.3 | 0.38× | 100.0 | 50.0 ± 16.7 | 0.50× | 95.3 | 73.0 ± 9.0 | 0.77× |
| PDRS | 100.0 | 100.0 ± 0.0 | 1.00× | 100.0 | 100.0 ± 0.0 | 1.00× | 95.3 | 100.0 ± 0.0 | 1.05× |
| PDRS | 100.0 | 100.0 ± 0.0 | 1.00× | 100.0 | 90.0 ± 10.0 | 0.90× | 95.3 | 100.0 ± 0.0 | 1.05× |
| PDRS | 100.0 | 43.2 ± 13.2 | 0.43× | 100.0 | 50.0 ± 16.7 | 0.50× | 95.3 | 53.8 ± 4.4 | 0.56× |

*Table 5.* Misspecification robustness ($n=10$). Top: single-modality models; bottom: MAVRL with full feedback set. Each row corrupts the data-side parameter of the named channel while the model assumes the well-specified value. Corrupted modality is blurred; e.g., PDRS denotes misspecified preferences. *Base.* = well-specified normalized return; *Misspec.* = mean±SEM; *Rat.* = Misspec./Base. Best per column in dark gray, second-best in light gray

.

dynamics.

**Grid-World Environments (grid_cliff, grid_sparse, grid_trap).** For the grid-world environments, we perturb the transition dynamics by introducing stochasticity in the agent's action execution. Specifically, we increase the probability of taking a random action $p_{\text{rand}} \in [0, 0.8]$, where $p_{\text{rand}} = 0$ corresponds to deterministic dynamics. These environments feature sparse rewards and punishing regions such as cliffs or traps, making robustness to action noise particularly important.

In the grid-world experiments, all methods converge to the same performance as $p_{\text{rand}} \to 0.8$. This behavior follows directly from how random actions are defined.

Let $a^*$ denote the action selected by a deterministic policy $\pi$, such that $\pi(a^* \mid s) = 1$. Under action noise with probability $p_{\text{rand}}$, the executed policy becomes

$$\pi_{\text{rand}}(a^* \mid s) = 1 - p_{\text{rand}}, \qquad \pi_{\text{rand}}(a' \mid s) = \frac{p_{\text{rand}}}{|\mathcal{A}| - 1} \quad \text{for } a' \neq a^*.$$

In the grid environments with $|\mathcal{A}| = 5$, setting $p_{\text{rand}} = 0.8$ yields $\pi_{\text{rand}}(a \mid s) = 0.2$ for all actions $a$, which is equivalent to a uniform random policy. Since returns are normalized such that the uniform policy achieves a value of $0.0$, all methods converge to this value at $p_{\text{rand}} = 0.8$.

For larger values of $p_{\text{rand}}$, unintended actions receive higher probability mass than the originally intended action, rendering deterministic action selection suboptimal. We therefore restrict our analysis to $p_{\text{rand}} \leq 0.8$.

**LunarLander-v3.** For LunarLander-v3, we perturb the dynamics by jointly increasing the magnitude of gravity and the strength of wind. Concretely, we vary the gravity parameter in $\{-10.0, -11.0, -12.0\}$, with default value $-10.0$, and the wind power in $[0.0, 25.0]$, with default value $0.0$. These perturbations induce increasingly challenging dynamics that require the agent to mitigate rapid descents caused by stronger gravity while compensating for lateral drift induced by wind.

**Acrobot-v1.** For Acrobot-v1, we perturb the system by varying the ratio between the two link lengths of the double pendulum. Increasing asymmetry between the links alters the inertia and coupling of the system, resulting in progressively different and more challenging dynamics compared to the unperturbed setting.

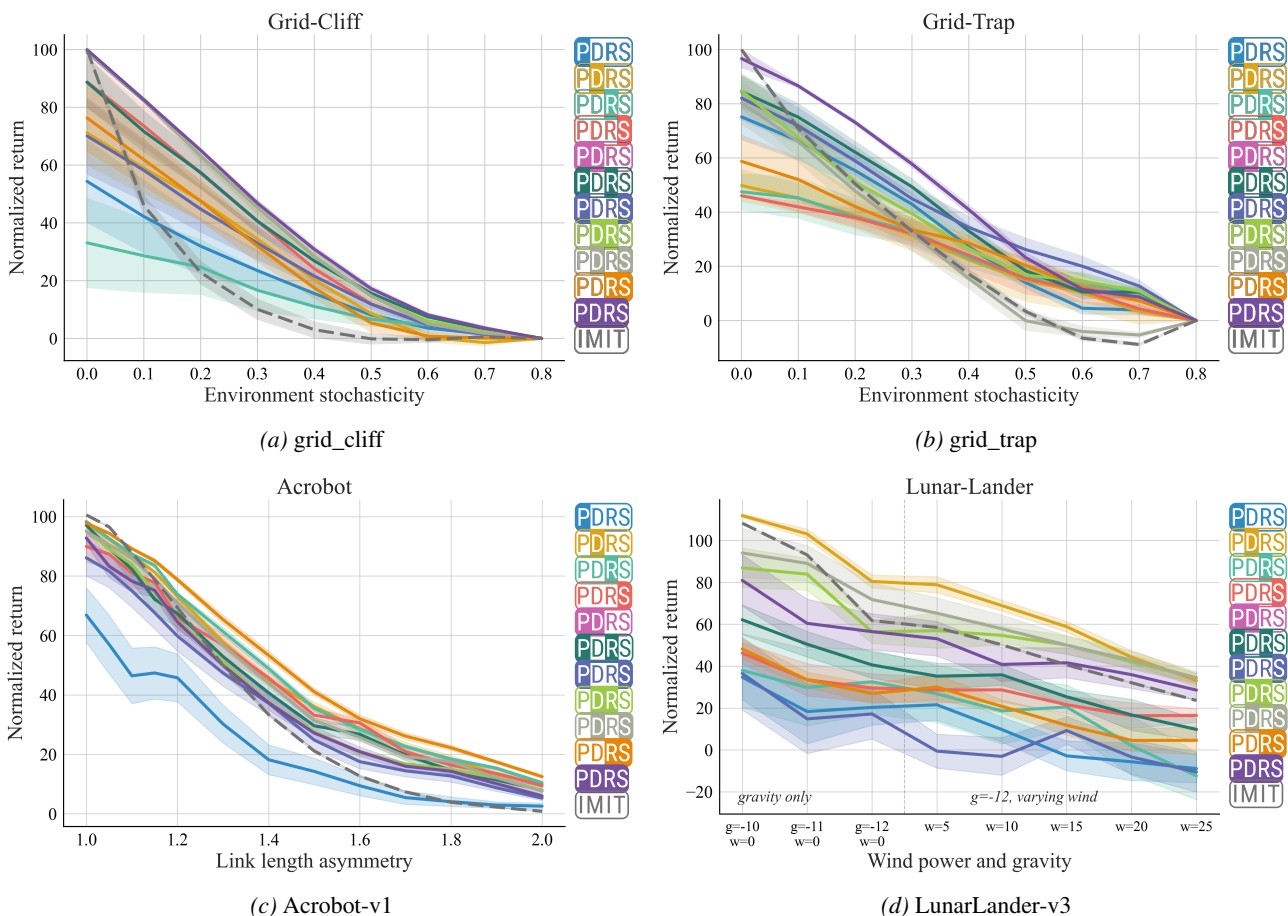

*(a)* grid_cliff

*(b)* grid_trap

*(c)* Acrobot-v1

*(d)* LunarLander-v3

*Figure 3.* Normalized mean returns ($n = 10$) of policies trained on rewards inferred by each method under three dynamics perturbation scenarios. Reward models and baselines are trained in the unperturbed setting and remain fixed throughout the variations. Returns are normalized such that 100.0 corresponds to the performance of an approximately optimal policy trained on the ground-truth reward *in the unperturbed setting* and, analogously, 0.0 corresponds to the performance of a uniformly random policy. Error bars denote standard error. (a) and (b) Increasing environmental stochasticity in grid_cliff and grid_trap. (c) Increasing ratio between pendulum handle lengths in Acrobot-v1. (d) Increasing gravity and wind-power in LunarLander-v3.

## F. Additional Qualitative Results

In the following section, we present additional qualitative results for all grid environments. Figures 4 to 6 each show two visualizations of the same inferred reward models: (a) a joint visual encoding of mean and variance, consistent with Figure 1, and (b) a separate visual encoding of mean and variance.

Note that we linearly normalize the inferred reward values per feedback type. This would not affect policy performance, since the induced behavior of the optimal policy is invariant to positive scaling and constant shifts of the reward function (Ng et al., 1999).

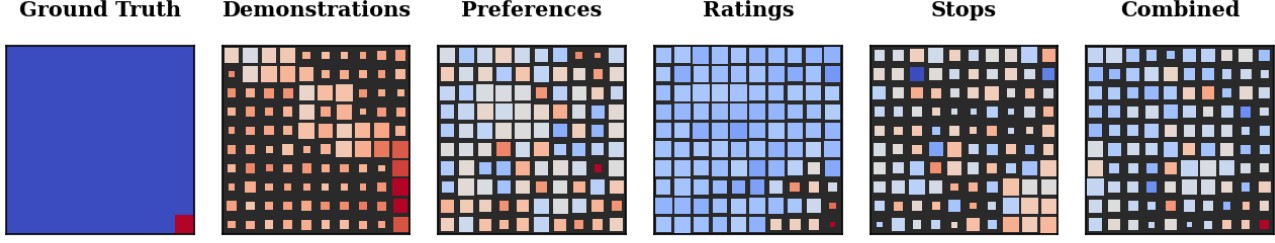

$\square$ *size = |1 - variance|,* $\square$ *color = mean value*

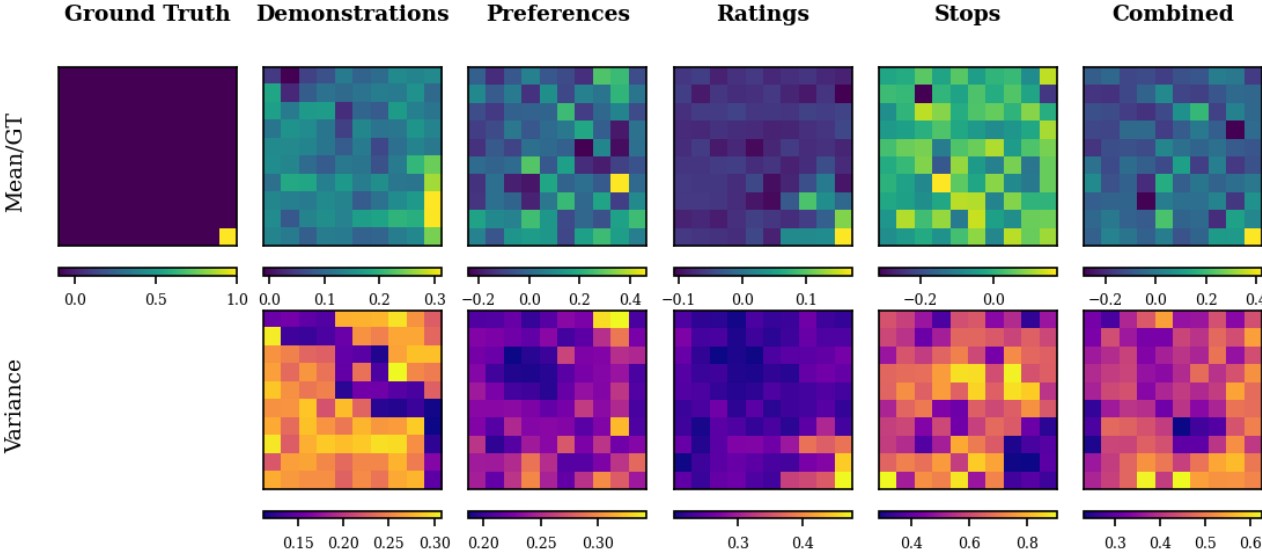

*Figure 4.* **grid_sparse**: Visualizations of inferred reward functions from 2 demonstrations, 256 pairwise comparisons, or 128 ratings on a $10 \times 10$ grid_sparse environment. The final column shows the result obtained when combining all feedback modalities. (a) Joint visual encoding of mean and variance. (b) Separate visual encoding of mean and variance for the same data. The rightmost column shows the result obtained when combining all feedback modalities.

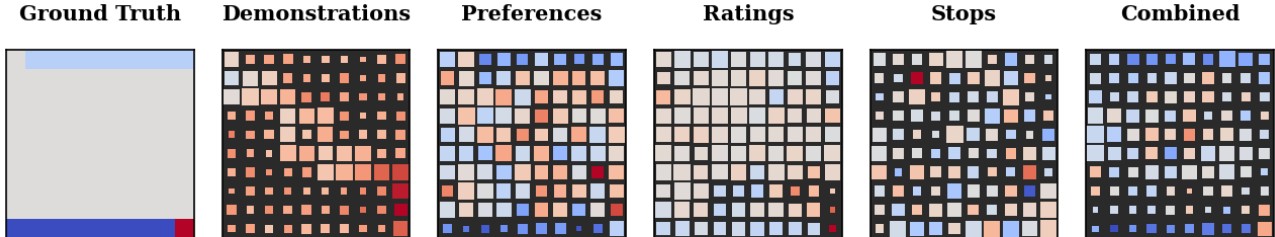

$\square$ *size = |1 - variance|*, $\square$ *color = mean value*

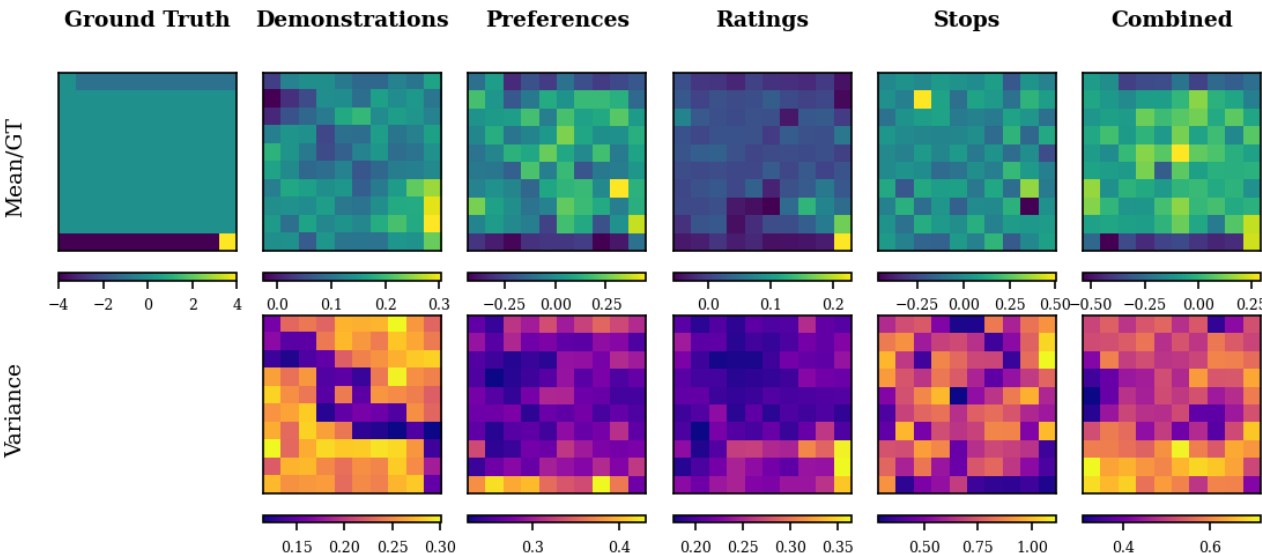

*Figure 5.* **grid_cliff**: Visualizations of inferred reward functions from 2 demonstrations, 256 pairwise comparisons, or 128 ratings on a $10 \times 10$ grid_cliff environment. The final column shows the result obtained when combining all feedback modalities. (a) Joint visual encoding of mean and variance. (b) Separate visual encoding of mean and variance for the same data. The rightmost column shows the result obtained when combining all feedback modalities.

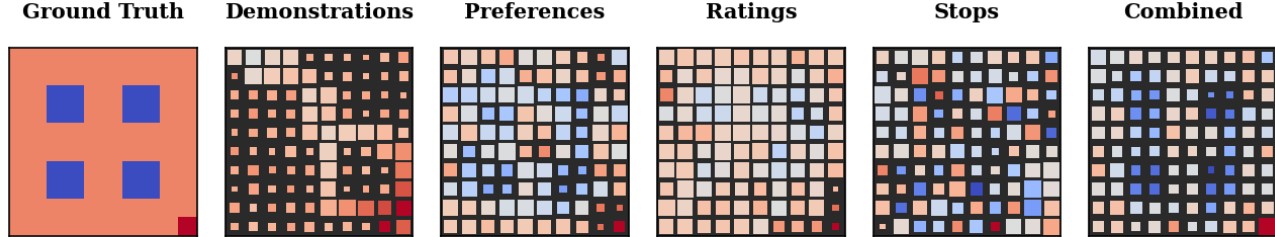

$\square$ *size = |1 - variance|,* $\square$ *color = mean value*

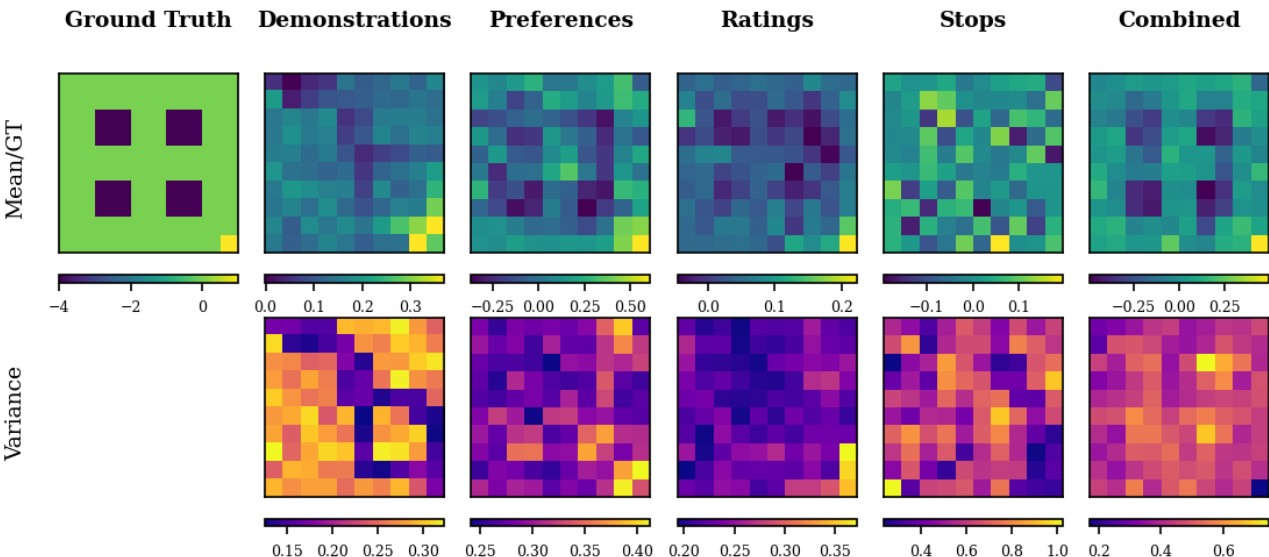

*Figure 6.* **grid_trap**: Visualizations of inferred reward functions from 2 demonstrations, 256 pairwise comparisons, or 128 ratings on a $10 \times 10$ grid_trap environment. The final column shows the result obtained when combining all feedback modalities. (a) Joint visual encoding of mean and variance. (b) Separate visual encoding of mean and variance for the same data. The rightmost column shows the result obtained when combining all feedback modalities.

