# OpenReview forum: "MAVRL: Learning Reward Functions from Multiple Feedback Types with Amortized Variational Inference"
_ICML.cc/2026/Conference — ICML 2026 regular_

### Official Review · Reviewer_A3Ho · 2026-03-03

**Soundness:** 3
**Presentation:** 3
**Significance:** 3
**Originality:** 2
**Overall Recommendation:** 5
**Confidence:** 4

**Summary:**

This paper addresses reward learning from multiple heterogeneous feedback types, such as demonstrations, pairwise preferences, scalar ratings, and stop interventions. Instead of optimizing manually weighted loss terms for each modality, the authors formulate the problem as Bayesian inference over a shared latent reward function, where each feedback type contributes through an explicit likelihood model.
They propose MAVRL, a scalable amortized variational inference algorithm that learns a shared variational reward encoder together with feedback-specific likelihoods under a single ELBO objective.
Experiments in grid-world and continuous-control environments show that different feedback modalities provide complementary information, and that jointly learning from multiple types often improves reward recovery and policy performance compared to single-modality baselines.

**Compliance With Llm Reviewing Policy:**

Affirmed.

**Final Justification:**

This paper proposes a principled Bayesian framework for learning reward functions from multiple heterogeneous feedback modalities. The formulation is technically sound, modular, and supported by experiments across tabular and continuous-control settings, along with useful diagnostic analyses.

My main concerns were whether the gains stem from true cross-modal complementarity rather than increased supervision, the need to specify likelihoods for each modality, and the limited analysis of performance degradation in some feedback combinations.

The rebuttal addresses these concerns to a meaningful extent. In particular, the matched-budget ablation convincingly shows that the improvements are not simply due to larger supervision, and the discussion of miscalibration provides a clearer explanation of failure cases.

Some limitations remain, notably the reliance on manually specified likelihoods for complex feedback and evaluation in relatively controlled settings, which may limit generality.

Overall, I find the paper technically solid and better supported after rebuttal. Therefore, I increase my overall assessment.

**Key Questions For Authors:**

1. The paper does not clearly specify how mini-batches are sampled across feedback modalities during training. Are equal batch sizes used per modality at each iteration? Or are samples drawn proportionally to dataset size? How is the relative contribution of each modality controlled in practice?
2. Demonstrations are primarily evaluated in relatively low-sample regimes, whereas other modalities are explored across wider budgets. Have you evaluated whether the complementarity claims hold when demonstrations are provided in larger quantities?
3. The framework requires specifying explicit probabilistic likelihoods for each feedback modality. How would the approach extend to more complex or unstructured feedback, such as natural language critiques or neural/LLM-based annotators? Do you envision learning these likelihoods jointly rather than specifying them manually?

**Limitations:**

yes

**Strengths And Weaknesses:**

**Strengths**
1. The paper presents a clear and principled Bayesian formulation for learning reward functions from multiple heterogeneous feedback types. By modeling each feedback modality as an explicit likelihood over a shared latent reward function, the work provides a coherent probabilistic interpretation that avoids heuristic loss weighting.

2. The proposed MAVRL algorithm extends amortized variational IRL to the multi-feedback setting in a technically sound way. The approach is modular and extensible in principle, allowing new feedback types to be incorporated by specifying corresponding likelihoods.

3. The experimental evaluation covers both tabular grid-world environments and continuous-control benchmarks. The results demonstrate that different feedback types induce different reward structures and that, in many settings, combining feedback improves robustness and overall performance.

**Weaknesses**
1. The framework requires specifying an explicit probabilistic likelihood for each feedback modality. While modular in principle, this may become challenging for complex or unstructured feedback (e.g., natural language critiques or neural annotators).

2. In the experimental comparisons, multi-feedback configurations (PDRS) use strictly more total feedback samples than single-modality baselines. While per-modality budgets are controlled, the total supervision differs. As a result, some of the observed gains may partially stem from increased overall supervision rather than purely from cross-modal complementarity. An ablation that matches total feedback budget across conditions would help isolate this effect.

3. Some experimental results show that certain feedback combinations underperform their constituent single-modality models. This suggests that joint inference can be sensitive to modality conflicts or noise misspecification. A deeper analysis of when and why such degradation occurs would strengthen the empirical claims.

---

> ### Author Rebuttal · Authors · 2026-03-31
>
> Thank you for taking the time to review our work and your interesting questions. We are glad the principled Bayesian formulation, modularity of MAVRL, and the breadth of the experimental evaluation were convincing to you. We address your concerns and questions below.
>
> ***
>
> ### (W1, Q3) Explicit likelihoods, unstructured feedback, and LLM-based annotators
> We agree that specifying explicit likelihoods is the main extensibility constraint of the current framework. That said, we believe this is less limiting in practice than it may appear.
>
> First, when LLM-based or neural annotators are used to elicit _structured_ feedback, ratings, preferences, or binary evaluations, the same likelihood models defined in this work apply directly, as is standard in the RL from AI Feedback literature, e.g., Lee et al. (2023), which uses the Bradley-Terry model for LLM judge preference labels. In this regime, MAVRL requires no modification.
>
> For _unstructured_ feedback such as natural language critiques, a grounding function (Jeon et al., 2020) is needed to map utterances to trajectories compliant with their semantics, which can then be treated as implicitly preferred. While learning such a grounding remains an open problem, capable neural annotators such as LLMs make it an increasingly tractable and promising direction for future work.
>
> ***
> ### (W2) Disentangling complementarity from total supervision
> This is an important point, and we have addressed it directly with an additional matched-budget ablation, now included in the paper. We use Bayesian Optimization to find strong feedback allocations under a fixed sample budget, and evaluate across a range of budget sizes. Key findings:
>
> |Sample Budget n|Pref.|Demos|Ratings|Stops|PDRS|Allocation ($n_p$, $n_d$, $n_r$, $n_s$)|
> |-|-|-|-|-|-|-|
> |*grid_trap*|||||||
> |8|46.1|55.7|42.7|-31.1|**60.7**|(4, 1, 0, 2)|
> |16|51.7|55.7|56.0|-31.1|**60.7**|(4, 1, 0, 2)|
> |32|55.6|55.7|**67.6**|34.6|62.0|(2, 1, 14, 14)|
> |64|79.2|55.7|**94.6**|40.5|89.5|(4, 0, 52, 2)|
> |128|**100.0**|55.7|**100.0**|50.2|94.6|(98, 0, 22, 6)|
> |256|**100.0**|55.7|**100.0**|85.3|**100.0**|(116, 0, 110, 6)|
> |*LunarLander-v3*|||||||
> |64|-4.7|**108.5**|22.0|19.0|38.6|(0, 16, 32, 16)|
> |128|-6.6|**108.5**|25.4|15.9|36.7|(64, 9, 32, 0)|
> |256|22.5|**108.5**|17.4|28.4|83.0|(0, 21, 144, 80)|
> |512|17.0|**108.5**|5.3|31.2|75.0|(0, 23, 144, 48)|
> |1024|29.7|**108.5**|17.0|44.6|74.3|(0, 19, 1120, 240)|
>
> Demo performance saturated at low $n_d$ (1 and 64 respectively) and further demonstrations degraded performance, so we report only the best-performing demo allocation.
>
> The combination of all feedback types (PDRS) achieves the best or second-best performance at every budget level, with gains most pronounced in low-data regimes, which indicates that this is where feedback complementarity matters most. As budgets grow and performance saturates, the marginal benefit diminishes naturally, consistent with complementarity rather than a supervision-size artifact.
> On LunarLander-v3, demonstrations dominate at all budgets, yet PDRS consistently outperforms the remaining single modalities without relying solely on them.
>
> ***
> ### (W3) Performance degradation in some feedback combinations
> We agree this deserves a more careful diagnosis. The evidence points to environment-specific miscalibration as the key cause and mechanism (distinct from misspecification). Even when the correct likelihood family is assumed, poorly chosen parameters (e.g., stop propensity too low) can cause catastrophic reward misidentification. Stop-containing combinations in CartPole-v1 likely suffered for this reason, as likelihood parameters were not tuned per environment. While our misspecification experiments (see response to Reviewer yqvk, W1/Q2) show MAVRL can partially compensate for corrupted modalities, miscalibration is harder to recover from when the affected modality dominates. Per-environment calibration, or learning likelihood parameters as latent variables, is the natural remedy and a clear direction for future work.
>
> ***
> ### (Q1) Mini-batch sampling across modalities
> At each iteration, we sample equally sized mini-batches from each modality and sum the log-likelihoods before backpropagation. When datasets differ in size, this implicitly oversamples smaller datasets, which can be viewed as importance weighting that prevents larger datasets from dominating the gradient signal. We have clarified this in the paper.
>
> ***
> ### (Q2) Complementarity at larger demonstration budgets
> Our choice to use fewer demonstrations reflects three considerations: (1) expert demonstrations are typically scarce in practice; (2) a single trajectory provides dense state-action supervision, making it more information-rich per sample than other feedback types; and (3) we empirically observe diminishing returns from additional demonstrations in sparse-reward settings, as shown in the previous table.
>
> ***
>
> Please let us know if any of your concerns remain, and we welcome any further questions.

---

> > ### Author Rebuttal · Reviewer_A3Ho · 2026-04-03
> >
> > Thank you for the detailed and constructive rebuttal. I appreciate the additional experiments and clarifications.
> >
> > In particular, the matched-budget ablation directly addresses my concern regarding whether the observed gains stem from increased total supervision or true cross-modal complementarity. The results and accompanying analysis provide convincing evidence that the improvements are not merely due to larger feedback budgets, which strengthens the empirical claims of the paper.
> >
> > I also found the discussion on performance degradation helpful, as it offers a more concrete explanation of failure modes beyond generic noise or misspecification.
> >
> > While some limitations remain, such as the need to specify likelihoods for complex feedback and the lack of large-scale evaluations, the rebuttal significantly improves my confidence in the empirical validation and the underlying mechanism.
> >
> > Based on these clarifications, I am inclined to increase my score.

---

> > > ### Author Response · Authors · 2026-04-03
> > >
> > > We are glad to hear that the additional experiments and clarifications were helpful, and we appreciate that you have reflected this in your score. Thank you for your thorough and constructive engagement throughout the discussion phase, as your comments led to meaningful improvements to the paper.

---

### Official Review · Reviewer_ocrk · 2026-03-11

**Soundness:** 3
**Presentation:** 3
**Significance:** 2
**Originality:** 3
**Overall Recommendation:** 4
**Confidence:** 2

**Summary:**

The paper considers reward learning from multiple feedback and frames it as Bayesian inference over a shared latent reward function. Further, an amortized variational inference algorithm, MAVRL, is proposed that optimizes a unified objective across different feedback types. Based on the empirical results, the MAVRL improves reward coverage, downstream policy performance, and robustness.

**Compliance With Llm Reviewing Policy:**

Affirmed.

**Key Questions For Authors:**

- How will MAVRL perform in large-scale and complex environments?
- How is the robustness to the feedback noise?

**Limitations:**

yes.

**Strengths And Weaknesses:**

Strengths:

- Formulating the reward learning from multiple feedback types as Bayesian inference over a shared latent reward function provides a new and interesting perspective.

- The proposed algorithm, MAVRL, provides extensible and efficient training with type-specific feedback datasets.

Weakness:

- As the author mentioned, it would be better to evaluate the proposed algorithm on a real-world dataset to verify the significant impact.

---

> ### Author Rebuttal · Authors · 2026-03-31
>
> Thank you for your feedback and helpful comments. We are glad you found the Bayesian formulation a novel and interesting perspective, and highlighted the potential of MAVRL's extensibility and training efficiency. We address your comments below.
>
> ---
>
> ### (W1) Evaluation on real-world datasets
>
> We agree that validating MAVRL on real human feedback would be an exciting next step. However, this is beyond the scope of the current work, which focuses on proposing a principled, heuristic-free algorithm for joint reward learning from heterogeneous feedback types. We consider this a natural and promising direction for future work.
>
> ---
>
> ### (Q1) Scalability to large-scale and complex environments
>
> MAVRL directly inherits the scalability benefits of the amortized variational framework, which has been shown to scale well to complex domains. Our continuous control experiments (CartPole, Acrobot, LunarLander) already demonstrate this in environments with continuous state-action spaces.
>
> We do not foresee fundamental obstacles to scaling further. The expressiveness of the encoder and decoder modules is the only relevant bottleneck, and standard choices (e.g., LLM-based encoders for language feedback) are a natural fit. We leave larger-scale evaluations to future work.
>
> ---
>
> ### (Q2) Robustness to feedback noise
>
> Following your suggestion, and the related question raised by Reviewer yqvk, we conducted a systematic investigation of feedback noise and model misspecification under equal feedback budget. In brief, MAVRL demonstrates an interesting compensation effect, where when one feedback type is misspecified, the remaining reliable modalities partially recover performance. We consider four misspecification types:
> - underestimated labeler noisiness for preferences ($\beta_\text{data}{=}0.01$, $\beta_\text{model}{=}2.5$) and demonstrations ($\beta_\text{data}{=}1.0$, $\beta_\text{model}{=}10.0$)
> - noisy ratings (Gaussian noise $\varepsilon \sim \mathcal{N}(0,5)$ added to utilities before label assignment);
> - miscalibrated stop propensity ($c_\text{data}{=}0.1$, $c_\text{model}{=}2.0$).
> We provide an excerpt of the new results below, as we cannot include plots in the rebuttal. The overall trend and observations are persistent across our results. Below, an excerpt for grid_trap (normalized mean reward):
>
> | Modality               | Correct spec. | Misspec. | Ratio    |
> | ---------------------- | ------------- | -------- | -------- |
> | Preferences            | 86.8          | 53.7     | 0.61     |
> | Demo                   | 62.0          | 45.4     | 0.73     |
> | Stop                   | 54.1          | 29.2     | 0.54     |
> | Ratings                | 67.9          | 51.5     | 0.76     |
> | PDRS (all misspec.)    | 67.6          | 46.4     | 0.69     |
> | PDRS (Pref. misspec.)  | 67.6          | 60.7     | **0.90** |
> | PDRS (Demos misspec.)  | 67.6          | 48.3     | 0.71     |
> | PDRS (Stops misspec.)  | 67.6          | 54.0     | 0.80     |
> | PDRS (Rating misspec.) | 67.6          | 60.1     | **0.90** |
>
> An interesting finding is that MAVRL seems to be able to compensate for noisy modalities: When one modality is misspecified, the remaining reliable signals substantially recover performance (degradation ratios 0.80–0.90 vs. 0.54–0.61 for single-modality baselines). When all modalities are simultaneously misspecified, MAVRL degrades no worse than single-modality methods, indicating no additional brittleness from joint inference.
>
> ---
>
> We hope this addresses your concerns and we welcome any further questions.

---

> > ### Author Rebuttal · Reviewer_ocrk · 2026-04-03
> >
> > Thanks for the detailed response. My questions have been addressed and the additional experiments help to clarify my concerns. I will maintain my original score.

---

> > > ### Author Response · Authors · 2026-04-03
> > >
> > > We are glad our responses were helpful and addressed your concerns. Thank you for engaging with us during the discussion phase and for your constructive comments.

---

### Official Review · Reviewer_8E4j · 2026-03-12

**Soundness:** 3
**Presentation:** 4
**Significance:** 2
**Originality:** 2
**Overall Recommendation:** 4
**Confidence:** 4

**Summary:**

This work investigates how heterogeneous reward signals can be jointly utilized to learn a robust reward function. The authors formulate this as a Bayesian inference problem, which is optimized using amortized variational inference. Finally, the paper demonstrates that learning from multiple feedback types simultaneously allows the algorithm to exploit complementary information, resulting in enhanced policy robustness.

**Compliance With Llm Reviewing Policy:**

Affirmed.

**Final Justification:**

The authors have addressed my concerns; hence, I am keeping my score.

**Key Questions For Authors:**

- Regarding the Demonstrations likelihood: should the expression for the probability of a trajectory given a reward function, $P(\xi| R)$, not explicitly account for the environment dynamics?

**Limitations:**

Yes

**Strengths And Weaknesses:**

## Strengths:
- The proposed Bayesian formulation eliminates the need for manual loss balancing across feedback types.

- Experiments across benchmarks show that combining feedback types generally outperforms single-type baselines. In particular, policies trained on multi-type feedback are more robust to environment dynamics perturbations compared to imitation learning or single-modality models.

## Weaknesses:
- The evaluation primarily compares the proposed multi-modal framework against its own single-modality baselines and a behavioral cloning (IMIT) baseline. While the results demonstrate clear benefits to policy robustness and reward recovery, the scope is limited to these baselines. Including comparisons against non-amortized Bayesian reward learning methods or other non-variational joint-learning heuristics could further validate whether the amortized variational approach is the most effective choice for heterogeneous signals.

---

> ### Author Rebuttal · Authors · 2026-03-31
>
> Thank you for taking the time to review our work. We are glad you found the Bayesian formulation effective in eliminating manual loss balancing, and that the multi-modal robustness results came through clearly. We address your comments below.
>
> ---
>
> ### (W1) Additional baselines
>
> Following your suggestion, we added two baselines to the paper, comparing both task performance and computational cost across 3 grid environments and the most challenging continuous control environment.
>
> 1. **MCMC:** a principled non-amortized Bayesian reward learning method representing the gold standard for posterior inference without variational approximations.
> 2. **Post-Hoc Reward Averaging:** a non-variational heuristic that combines heterogeneous reward estimates through simple ensemble averaging after independent training (cf. Metz et al., 2025).
>
> | Environment    | Post-Hoc Avg | MCMC        |                | MAVRL       |               |
> | -------------- | ------------ | ----------- | -------------- | ----------- | ------------- |
> |                | Performance  | Performance | Wall-Time      | Performance | Wall-Time     |
> | grid_sparse    | **100.0 ± 0.0**  | **100.0 ± 0.0** | 154.4 ± 0.7  s | **100.0 ± 0.0** | **4.5 ± 0.0 s**   |
> | grid_cliff     | 55.6 ± 14.4  | **100.0 ± 0.0** | 156.0 ± 0.4  s | 73.4 ± 13.3 | **4.9 ± 0.4 s**   |
> | grid_trap      | 73.0 ± 9.0   | **93.0 ± 7.0**  | 155.0 ± 0.4  s | 91.5 ± 5.5  | **4.6 ± 0.0 s**   |
> | LunarLander-v3 | 36.7 ± 2.6   | —           | —              | **64.0 ± 8.6**  | **46.5 ± 10.1 s** |
>
> These comparisons make two distinct arguments.
>
> Compared to **MCMC**: MAVRL achieves comparable or superior performance at a fraction of the computational cost (~30× faster on grid environments), and unlike MCMC does not require an inner-loop MDP solve. Note that MCMC is intractable for continuous control settings, whereas MAVRL easily scales to these domains.
>
> Compared to **Post-Hoc Reward Averaging**: naïve averaging is an inadequate substitute for principled joint learning. Since reward models are trained independently, post-hoc averaging is blind to cross-modal complementarity and conflates signals with different scales and reward shaping, yielding poorly shaped estimates for downstream policy training.
>
> Together, these results strengthen the case for MAVRL on two fronts: it is both **effective** (vs. Post-Hoc Avg) and **efficient** (vs. MCMC).
>
> ---
>
> ### (Q1) Demonstration likelihood and environment dynamics
>
> The Boltzmann-rational likelihood is parameterized by Q-values, which implicitly encode the environment dynamics. Explicitly accounting for them is therefore unnecessary. Ordinarily, computing Q-values from $R$ requires solving the MDP in an inner loop, but, importantly, MAVRL sidesteps this by learning a Q-model jointly with $R$, maintaining mutual consistency throughout training via TD-error regularization.
>
> ---
>
> We hope these additions address your concerns and we welcome any further questions.

---

> > ### Author Rebuttal · Reviewer_8E4j · 2026-04-03
> >
> > I appreciate the authors adding these baselines as they provide needed framing against related reward learning methods. This addresses my previous concerns, and I am happy to keep my current score.

---

> > > ### Author Response · Authors · 2026-04-03
> > >
> > > Thank you for acknowledging that your concerns have been addressed. We appreciated your engagement during the discussion phase and your helpful pointers and positive feedback.

---

### Official Review · Reviewer_yqvk · 2026-03-12

**Soundness:** 3
**Presentation:** 3
**Significance:** 3
**Originality:** 3
**Overall Recommendation:** 4
**Confidence:** 3

**Summary:**

This paper consider reward learning from diverse human feedback and formulate this as bayesian inference over a common latent reward function. Additionally, it presents MAVRL, which optimizes a single evidence lower bound to jointly learn a variational reward encoder and feedback-specific likelihood decoders without the need for manual loss weighting across modalities.

**Compliance With Llm Reviewing Policy:**

Affirmed.

**Final Justification:**

The paper proposes a principled Bayesian formulation (MAVRL) for jointly learning reward functions from heterogeneous feedback types via a single ELBO, with each modality contributing through its own likelihood. The stop-feedback hazard model is a genuine novelty, and the framework's extensibility is a clear architectural merit.

My main concerns were: (1) lack of robustness evaluation under feedback model misspecification, (2) overstated elimination of loss balancing given that $\lambda_{\text{KL}}$, $\lambda_{\text{TD}}$, and noise parameters still require tuning, and (3) insufficient diagnosis of stop-feedback degradation in CartPole.

The rebuttal substantially addressed (1) with new misspecification experiments showing that MAVRL degrades gracefully when a single modality is misspecified (0.80–0.90 retention ratio vs. 0.54–0.61 for single-type baselines), a finding that itself strengthens the paper's complementary-feedback narrative. For (2), the authors helpfully distinguished annotator-behavior parameters from cross-modality loss weights, though the practical need to tune $\lambda_{\text{KL}}$ and $\lambda_{\text{TD}}$ remains. For (3), linking CartPole stop-feedback degradation to $\lambda$ miscalibration is plausible but still somewhat surface-level.

The core contribution replacing ad-hoc loss combination with likelihood-based Bayesian integratio is sound and fills a genuine gap. Remaining limitations (synthetic-only evaluation, stop-feedback calibration sensitivity, low-dimensional environments) are acknowledged and do not undermine the conceptual advance. I maintain my score.

**Key Questions For Authors:**

Q1: All feedback-specific NLL terms must enter the ELBO with equal weight in order for the claim of eliminating manual loss balancing to be true. However, the noise-model parameters ($\beta$, cutpoints, $\lambda$, $\rho$) within each likelihood also influence the relative contribution of each feedback type, and $\lambda_{\text{KL}}$ and $\lambda_{\text{TD}}$ still need to be set. How were these selected?

Q2: All feedback types in the synthetic feedback setup are generated in a way that perfectly complies with the likelihood assumptions of MAVRL. Have you examined the effects of feedback that deviates from these presumptions, such as when systematic biases in ratings, inconsistent pairwise preferences, or delayed stops are introduced?

Q3: A number of stop-containing feedback combinations in CartPole perform significantly worse than single-type baselines (PDRS at 12.2 vs. PDRS at 69.8). Conflicts between gradients derived from feedback are your explanation for this. Could you give a more specific diagnosis? Do the stop-derived likelihood gradients, for example, influence the reward encoder in a way that deviates from ratings or demonstrations?

**Limitations:**

yes

**Strengths And Weaknesses:**

Strengths: The Bayesian formulation avoids the need to collapse heterogeneous signals into a common representation by allowing each type of feedback to contribute to a shared reward posterior through its own likelihood. The stop feedback likelihood is novel and closes a practical gap by modeling it as a hazard over cumulative suboptimality. In reward-uncertainty space, Figure 1 clearly illustrates how different types of feedback complement one another.



Weaknesses: Robustness to human biases and model misspecification is not tested because all experiments use low-dimensional environments with synthetic feedback precisely aligned to the likelihood assumptions of the model. Since \lambda_KL, lambda_TD, and likelihood noise parameters still need to be tuned, the claim that loss balancing has been eliminated is exaggerated. In CartPole, performance degradation from specific feedback combinations is observed but not sufficiently diagnosed.

---

> ### Author Rebuttal · Authors · 2026-03-31
>
> Thank you for your careful reading and constructive feedback. We are glad you found the Bayesian formulation principled, the stop-feedback likelihood novel, and Figure 1 effective. We address the weaknesses and questions below.
>
> ---
>
> ### (W1, Q2) Robustness to feedback model misspecification
> We agree that adding an evaluation of the effect of feedback model specification will strengthen our contribution. Following your suggestion, we conducted experiments to address this question. We provide an excerpt of the results for grid_trap (mean normalized return) below and added the full plots to the paper. The overall trend and observations are persistent across our results. We consider four misspecification types with specific settings for the example results in parentheses:
>
> - underestimated **labeler noisiness** for preferences ($\beta_\text{data}{=}0.01$, $\beta_\text{model}{=}2.5$) and demonstrations ($\beta_\text{data}{=}1.0$, $\beta_\text{model}{=}10.0$)
> - **noisy ratings** (Gaussian noise $\varepsilon \sim \mathcal{N}(0,5)$ added to utilities before label assignment);
> - **miscalibrated stop propensity** ($c_\text{data}{=}0.1$, $c_\text{model}{=}2.0$).
>
> | Modality | Correct spec. | Misspec. | Ratio |
> | --- | --- | --- | -- |
> | Preferences | 86.8  | 53.7     | 0.61     |
> | Demo | 62.0  | 45.4     | 0.73     |
> | Stop | 54.1 | 29.2     | 0.54     |
> | Ratings | 67.9   | 51.5     | 0.76     |
> | PDRS (all misspec.)    | 67.6   | 46.4     | 0.69     |
> | PDRS (Pref. misspec.)  | 67.6   | 60.7     | **0.90** |
> | PDRS (Demos misspec.)  | 67.6          | 48.3     | 0.71     |
> | PDRS (Stops misspec.)  | 67.6          | 54.0     | 0.80     |
> | PDRS (Rating misspec.) | 67.6          | 60.1     | **0.90** |
>
> An interesting finding is that MAVRL seems to be able to compensate for noisy modalities: when one modality is misspecified, the remaining reliable signals substantially recover performance (degradation ratios 0.80–0.90 vs. 0.54–0.61 for single-modality baselines). When all modalities are simultaneously misspecified, MAVRL degrades no worse than single-modality methods, indicating no additional brittleness from joint inference.
>
> ---
>
> ### (W2, Q1) Loss balancing, feedback model parameters, and regularizers
>
> We appreciate this clarification request and want to clarify the role of the different parameters and what loss balancing MAVRL eliminates.
>
> **Regularizers $\lambda_\text{KL}, \lambda_\text{TD}$:** These regularizers enforce posterior regularization and reward–Q-value consistency. We found performance trends to be largely stable across a broad range of values ($[0.125, 2.0]$), though Q-value-based feedback types (demos and stops) require non-zero $\lambda_\text{TD}$ by definition, and feedback combinations with more modalities tend to benefit from slightly higher values of both ($\lambda_\text{TD} = \lambda_\text{KL} = 2.0$). We tuned these via standard model selection. We updated the paper to explicitly mention the effect of these regularization coefficients.
>
> **Likelihood-specific noise parameters ($\beta, \rho, c, \ldots$):** These parameters characterize annotator behavior, rationality, tolerance to suboptimality, and rating granularity, grounded in well-established models of human feedback. Our goal is not to advance feedback modeling per se, but to provide a principled approach to jointly learn from diverse feedback given such models. It is worth noting that MAVRL does support estimating these noise parameters from data by treating them as additional latent variables inferred under behaviorally plausible priors.
>
> **What MAVRL actually eliminates:** The core claim is that MAVRL removes the need to explicitly balance how much each feedback type contributes to the reward estimator. To make this concrete: given 1 demonstration and 25 preferences with known noise parameters $\beta_{\text{pref}}$ and $\beta_{\text{demo}}$, what weight should each receive in a combined loss or a post-hoc reward ensemble? For general MDPs, any answer is essentially arbitrary. MAVRL sidesteps this question entirely. All feedback enters the ELBO on equal footing, with relative influence determined by the data and the likelihood models rather than manual tuning. We agree that we should be more explicit about this and have sharpened this claim in the paper.
>
> ---
>
> ## (W3, Q3) Stop feedback degradation in CartPole
>
> We agree this calls for a clearer diagnosis. Our additional misspecification experiments confirm that stop feedback performance is sensitive to the calibration of the propensity parameter $c$. This is consistent with the theoretical model: when $c$ is miscalibrated, the model can misinterpret suboptimal transitions. Per-environment calibration of $c$, or learning it as a latent variable, is a natural direction for improvement, now discussed in the paper.
>
> ---
>
> We hope these clarifications and additional results address your concerns. We welcome any remaining questions.

---

> > ### Author Rebuttal · Reviewer_yqvk · 2026-04-02
> >
> > Thank you to the authors for their diligence in conducting extra experiments and offering detailed explanations. These additions have cleared up most of my questions. As such, I am maintaining my score.

---

> > > ### Author Response · Authors · 2026-04-03
> > >
> > > We are happy to hear that we could resolve your concerns. Thank you for engaging with us during the discussion phase and for your helpful pointers, which strengthened the paper.

---

### Decision · Program_Chairs · 2026-04-30

**Decision:**

Accept (regular)

**Comment:**

Reviewers agreed that the paper presents a principled Bayesian formulation for learning reward functions from multiple heterogeneous feedback types. Reviewers highlighted three main reasons for acceptance: (1) the method is technically solid and modular, with a unified ELBO objective that avoids manual loss balancing across modalities; (2) the author response strengthened the paper substantially by adding matched-budget and misspecification experiments, which support the claim that the gains come from true cross-modal complementarity rather than simply more supervision; and (3) the empirical results are convincing overall, showing improved reward recovery, policy robustness, and scalability across grid-world and continuous-control settings.